

# Predicting evapotranspiration from drone-based thermography – a method comparison in a tropical oil palm plantation

Florian Ellsäßer[1], Christian Stiegler[2], Alexander Röll[1], Tania June[3], Hendrayanto[4], Alexander Knohl[2,5], Dirk Hölscher[1,5]

[1] University of Goettingen, Tropical Silviculture and Forest Ecology, Büsgenweg 1, 37077 Göttingen, Germany

[2] University of Goettingen, Bioclimatology, Büsgenweg 2, 37077 Göttingen Germany

[3] Bogor Agricultural University, Geophysics and Meteorology, Jln. Meranti, 16680 Bogor, Indonesia

[4] Bogor Agricultural University, Forest Management, Kampus IPB Darmaga, 16680 Bogor, Indonesia

[5] University of Goettingen, Centre of Biodiversity and Sustainable Land Use, Platz der Göttinger Sieben 5, 37073 Göttingen, Germany

*Correspondence to:* Florian Ellsäßer (fellsae@gwdg.de)



## Abstract

For the assessment of evapotranspiration, near-surface airborne thermography offers new opportunities for studies with high numbers of spatial replicates and in a fine spatial resolution. We tested drone-based thermography and the subsequent application of three energy balance models (DATTUTDUT, TSEB-PT, DTD) using the widely accepted eddy covariance technique as a reference method. The study site was a mature oil palm plantation in lowland Sumatra, Indonesia. For the 61 flight missions, latent heat flux estimates of the DATTUTDUT model with measured net radiation agreed well with eddy covariance measurements ($r^2$=0.85; MAE=47; RMSE=60) across variable weather conditions and daytimes. Confidence intervals for slope and intercept of a model II Deming regression suggest no difference between drone-based and eddy covariance method, thus indicating interchangeability. TSEB-PT and DTD yielded agreeable results, but all three models are sensitive to the configuration of the net radiation assessment. Overall, we conclude that drone-based thermography with energy-balance modeling is a reliable method complementing available methods for evapotranspiration studies. It offers promising, additional opportunities for fine grain and spatially explicit studies.



## 1 Introduction

Evapotranspiration (ET) is a central flux in the hydrological cycle on a regional and on a global scale. Terrestrial ET consumes almost two-thirds of terrestrial precipitation (Oki and Kanae, 2006). There is an interest in better understanding ET and its drivers as climate change is expected to increase atmospheric evaporative demand and droughts are predicted to become more severe and frequent in the future (Prudhomme et al., 2014). ET is also strongly affected by land-cover and land-use changes, which are currently very pronounced in tropical regions (Hansen et al., 2013).

The eddy covariance technique (EC) is a widely accepted and well-established method to quantify ET at the stand scale (Baldocchi et al., 2001; Fisher et al., 2017). It results in a single latent heat flux (LE) value integrated over the footprint of the EC tower at a given time that can be converted to an ET estimate. A spatial fine grain attribution of different surface patches to this overall ET value is generally not possible. The EC method is costly and labor intensive, and therefore, a relatively low number of spatial replicates within a given region and among its different ecosystems are typically available. The EC method also has certain constrains regarding topography, atmospheric turbulence and landscape heterogeneity (Göckede et al., 2008).

A complementary approach for assessing LE at larger spatial scales is the use of remotely sensed land surface temperatures (LST) as boundary conditions for energy balance modeling and subsequent conversion to ET (Brenner et al., 2017; Guzinski et al., 2014; Hoffmann et al., 2016; Ortega-Farías et al., 2016; Xia et al., 2016). Compared to the EC method, this approach can potentially increase the number of spatial replicates within and among ecosystems and is also applicable in challenging terrain. Remotely sensed LSTs are regarded as good indicators for plant water use, stress and transpiration (Jones and Vaughan, 2010). One approach to obtain LST data is the use of satellite-based observations (Allen et al., 2007; Bastiaanssen et al., 1998; Ershadi et al., 2013). However, the spatial resolution of satellite data such as Landsat TM, ASTER, MODIS or AVHRR ranges from 90 m to 1 km, limiting the distinction of plant canopies and soil (Berni et al., 2009). A higher temporal resolution of satellite-based thermal infrared (TIR) observations is usually associated with a lower spatial resolution, and TIR data from satellites in both high spatial and high temporal resolution are not yet available (Brenner et al., 2017). Additionally, clouds are barriers for thermal radiation and therefore have a strong effect on the quality and availability of satellite-based TIR observations (Guzinski et al., 2013). This is of particular importance in regions with frequent cloud cover such as in tropical environments.

An alternative, recently emerging approach to measure LSTs is the use of drones. Radiometric TIR sensors for LST recording have become light-weight and affordable, and drones are now capable of carrying adequate payloads for reasonable timespans. Near-surface thermography-based studies allow





temporal resolutions in flexible, e.g. hourly time steps and a spatial resolution in the decimeter scale or finer. Drone-based TIR recording and subsequent modeling of LE with energy balance models has previously shown promising results for short grass and crop vegetation in Central Europe (Brenner et al., 2018; Hoffmann et al., 2016). However, remote sensing of LST from drones is challenging and involves careful planning. Recording LST close to the surface results in a high resolution but reduces the area covered in a certain time span compared to surveying from a higher altitude. Increasing flight altitude reduces spatial resolution of LST images and thus increases the averaging of surface temperatures from individual canopies, soil patches and branches from neighboring canopies into a single pixel (Still et al., 2019). Further, air humidity can have a major effect on measurement accuracy as water vapor does not only attenuate the signals from the surface of interest to the sensor, but also emits its own thermal radiation (Still et al., 2019).

Different energy balance models are available to compute LE from LST and subsequently calculate ET. In the one-source energy balance model DATTUTDUT (Deriving Atmosphere Turbulent Transport Useful To Dummies Using Temperature) (Timmermans et al., 2015) fluxes are estimated by relating single pixel temperatures to local temperature extremes; the hottest and a group of coldest pixels in the image (Timmermans et al., 2015). Two-source energy balance models such as TSEB (Two-Source Energy Balance) (Norman et al., 1995) and DTD (Dual Temperature Difference) (Norman et al., 2000) divide measured LSTs into a vegetation and a soil fraction. Several adaptions of these models were developed; the TSEB-PT model as described in Hoffmann et al. (2016), uses the Priestley-Taylor coefficient (PT) to determine canopy H flux and subsequently calculate the other fractions from the surface energy balance. TSEB-PT is based on the temperature difference between LST and air temperature (Norman et al., 1995). Expanding this concept, DTD uses a dual-temperature difference from an additional early morning set of measurements to account for biases in remotely sensed LSTs (Hoffmann et al., 2016; Norman et al., 2000). Crucial in applying such energy balance models is how the net radiation (Rn) is implemented. In the original formulation of the DATTUTDUT model Rn is fully modeled, assuming a range of prerequisites and environmental conditions (Timmermans et al., 2015). TSEB-PT and DTD models use measured short and long-wave radiation to estimate Rn as a sum of in- and outgoing long- and short-wave radiation (Norman et al., 1995, 2000). Using airplanes or drones to record LSTs, the three models previously showed promising results for grass and crop surfaces in temperate and subtropical regions (Brenner et al., 2017, 2018; Hoffmann et al., 2016; Xia et al., 2016). However, the limited number of temporal replicates for a given study site constrained previous studies to using error terms and correlation coefficients. To our knowledge, a comprehensive method comparison considering potential errors in both reference method (e.g. the EC technique) and novel drone-based approaches is not yet available.

The presented study was conducted in the lowlands of Jambi province (Sumatra, Indonesia) where over the last decades, large areas of rainforest have been converted to rubber and oil palm plantations (Clough





et al., 2016; Margono et al., 2012). This resulted in regional-scale changes in transpiration (Röll et al.,
2019) and land surface warming (Sabajo et al., 2017). We assessed energy fluxes in a mature monoculture
oil palm plantation and compared the LE estimates of drone-based methods with the established EC
method as measured ground-based reference. Three energy-balance models (DATTUTDUT, TSEB-PT,
DTD) were tested, each with three different configurations for determination of Rn (fully modeling Rn,
Rn estimates based on short-wave irradiance and measuring Rn). The objectives of our study were to
compare LE estimates from the drone-based methods to the EC technique, with a special focus on the
detection of proportional and continuous errors among the methods and an evaluation of the models
prediction performance.

**2 Methods**

**2.1 Study site**

The study site is located in the lowlands of Jambi province (Sumatra, Indonesia) near the equator (E
103.3914411, N -1.6929879, 76 m a.s.l.). Average annual air temperature in the region is 26.5°C and
average annual precipitation is 2235 mm yr$^{-1}$ (Drescher et al., 2016). At the time of our measurement
campaign in August 2017, the studied monoculture oil palm (*Elaeis guineensis*) plantation was 15 years
old. Palm stem density was 140 palms ha$^{-1}$, with an average palm height of 14.3 m and an average canopy
radius of 4.5 m. Leaf area index (LAI) was estimated at 3.64 m$^2$ m$^{-2}$ (Fan et al., 2015) and canopy cover
was estimated to be 90%. Plantation management included the removal of older and non-vital leaves from
the oil palms, herbicide application to remove most understory plants and fertilization (196 kg N ha$^{-1}$ yr$^{-1}$
) (Meijide et al., 2017). The average annual oil palm yield is 27.7 Mg ha$^{-1}$. An EC tower (22 m height)
is situated in the center of the site with a fetch of up to 500 m in each direction (Meijide et al., 2017) (Fig.
136  1).



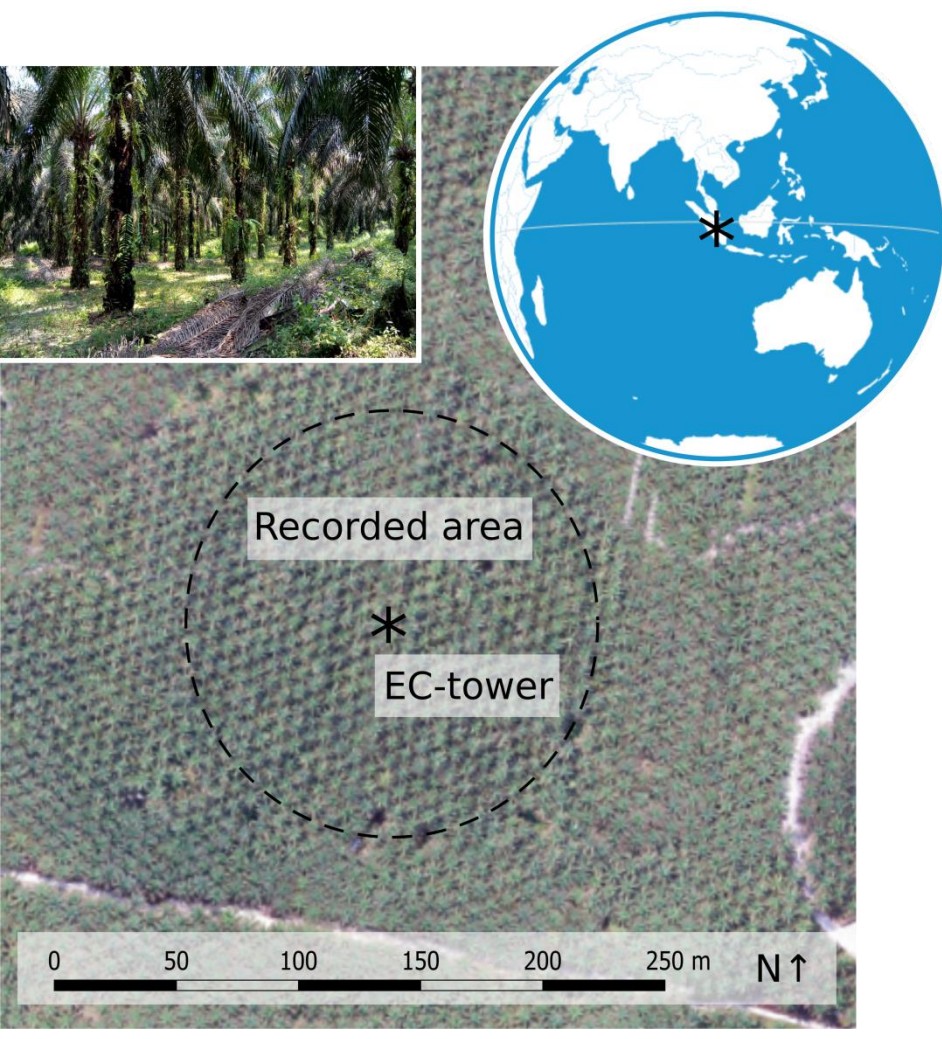

**Figure 1:** The study site in a mature commercial oil palm plantation in the lowlands of Jambi province, Sumatra, Indonesia.

## 2.2 Drone-based image acquisition

We used an octocopter drone (MK EASY Okto V3; HiSystems, Germany) equipped with a thermal and an RGB camera mounted in a stereo setup on a gimbal to ensure nadir perspective. The radiometric thermal camera was a FLIR Tau 2 640 (FLIR Systems, USA) attached to a TeAx Thermo-capture module (TeAx Technology, Germany). The sensor covers spectral bands ranging from 7.5 to 13.5 µm with a thermal accuracy of 0.04 K. The RGB camera was based on an Omnivision OV12890 CMOS-Sensor (Omnivision, USA) with a 170° FOV fish-eye lens. Instead of the mosaicking approaches applied in most





of the mentioned previous studies, we used a single image recording concept as faster image acquisition
allows for a denser temporal resolution of LSTs. To capture an area of 100 m radius around the EC tower
in a single shot of the thermal camera, images were taken from 260 m altitude. Image corners were
removed due to vignetting effects. During a consecutive five-day flight campaign in August 2017, 61 LST
data sets and matching EC measurements were recorded. Flights were conducted between 9 am and 4 pm
local time, in accordance with the 30 min intervals of the EC averaging cycles, resulting in 10 to 14 flights
per day (Table A1). All LSTs were measured using a fixed emissivity of one as the energy balance models
would introduce specific soil and vegetation emissivities in the process.

**2.3 Energy balance models**

LSTs are recorded as 'snapshots' representing an instantaneous state of surface temperatures. Soil-
Vegetation-Atmosphere Transfer models use these instantaneous observations of LST to solve the energy
balance equation and estimate instantaneous fluxes. In our study the one-source energy balance model
DATTUTDUT (Timmermans et al., 2015) and two two-source energy balance models, TSEB-PT
(Norman et al., 1995) and DTD (Norman et al., 2000), were applied.
The key input for the DATTUTDUT model is a LST map from which the hottest and coldest pixels are
extracted, assuming that hot pixels are a result of very little to no ET and cold pixels origin from high ET
(Timmermans et al., 2015). Apart from the LST map, all further inputs for the model such as albedo,
atmospheric transmissivity and surface emissivity are either calculated from temperature extremes in the
map, location and time or are fixed values as suggested in Timmermans et al. (2015). A version of the
DATTUTDUT model has recently been implemented in form of a QGIS3 plugin (QWaterModel) with
focus on easy usability (Ellsäßer et al., in press). TSEB-PT calculates surface-energy budgets from the
recorded LSTs splitting observations into a canopy and a soil fraction (Norman et al., 1995; Song et al.,
2016; Xia et al., 2016). The Priestley-Taylor (PT) approximation is used to calculate the sensible heat
flux for the canopy fraction from net radiation divergence estimates (Hoffmann et al., 2016). With the
sensible heat flux known, canopy and soil temperature are calculated and with known resistances fluxes
are computed (Hoffmann et al., 2016). Calculation of aerodynamic temperature by using an excess
resistance term is not needed, since TSEB-PT uses directional radiometric temperature as input
(Hoffmann et al., 2016). The TSEB-PT model requires additional *in situ* meteorological measurements
of long- and short-wave radiation, wind speed, barometric pressure and relative humidity, which in our
case were recorded at the EC tower. Further, data on LAI as well as surface and canopy albedo are
required. In the DTD model, the absolute temperatures of land surface and air (as used in the TSEB-PT)
are supplemented with a second set of early morning reference measurements of LST and air temperature,
thus creating a dual-temperature difference (Norman et al., 2000). This relates measurements at any time
during the day to measurements recorded in the morning, when fluxes are assumed to be minimal, and





thereby accounts for measurement biases of LST (Anderson et al., 1997; Hoffmann et al., 2016). H flux
is then calculated using the time-differential temperature and a series resistance network as it is
recommended for densely vegetated regions to consider interaction of soil and canopy fluxes (Guzinski
et al., 2014; Li et al., 2005). We used two thermal cameras attached to the EC tower (see EC methodology
for details and Sect. 2.7 for the limitations) for the necessary early morning reference readings of absolute
temperature and used the averaged LSTs from the thermal images to create a uniform map as input for
the DTD model (similar as e.g. in Hoffmann et al., 2016). Ground heat flux (G) was computed in the same
way for all three models, i.e. as a linear function of Rn (Liebethal and Foken, 2007). More details on the
applied models are provided as supplementary information (Appendix II).
Modeled LE estimates are highly sensitive to the type of Rn estimates used. Consequently, in our study
we compared three different configurations to include Rn into each of the three mentioned energy balance
models: a) fully modeled Rn from sun-earth-geometrics (Rn_mod) as in the original procedure of the
DATTUTDUT model with no option to consider clouds, b) measuring only incoming short-wave
radiation and calculating net short-wave using the surface albedo, while net long-wave is calculated from
measured air temperature, LST, the Stefan Bolzmann equation and atmospheric emissivity to estimate Rn
(Rn_sw) as in Guzinski et al. (2013), and c) measuring the four components of the radiation budget
independently and calculating Rn (Rn_mes), as is the case for the original procedure for the TSEB-PT
and DTD model and the reference EC method. From the presented results (LE flux densities normalized
by area, in W m$^{-2}$), we further calculated hourly ET rates (amount of water, mm h$^{-1}$); in this, we assumed
a stable relationship of the fluxes during the estimation period (Cammalleri et al., 2014), and followed
Timmermans et al. (2015) for calculating the latent heat of vaporization.

**2.4 Eddy covariance measurements**

The micrometeorological tower is located in the center of the study site (Fig. 1). The EC technique was
used to measure LE and H fluxes from high frequency (10 Hz) measurements of above-canopy water
vapor concentration, sonic temperature, and 3-D wind components. The flux system consisted of a sonic
anemometer (Metek uSonic-3 Scientific, Elmshorn, Germany) and a fast response open-path $CO_2$/$H_2O$
infrared gas analyzer (Li-Cor7500A, LI-COR Inc. Lincoln, USA) installed at 22 m height. Meteorological
variables were measured every 10 sec, averaged to 10 min means and stored on a DL16 Pro data logger
(Thies Clima, Göttingen, Germany). Rn and its components were measured with a net radiometer (CNR4,
Kipp & Zonen, Delft, The Netherlands) at 22 m height. Air temperature and relative humidity were
measured with thermohygrometers (type 1.1025.55.000, Thies Clima, Göttingen, Germany) at 22 m
height. Further, a wind direction sensor (Thies Clima, Göttingen, Germany) (22 m height) and 3-cup
anemometers (Thies Clima, Göttingen, Germany) (18.5, 15.4, 13, and 2.3 m height) for wind speed
measurements were installed on the tower. Two fixed thermal cameras (IR100 Radiometer, Campbell





Scientific Inc., Logan, USA) on top of the tower (22 m height) were used for early morning measurements
of LST. Ground heat flux was measured using heat flux plates (HFP01, Huxeflux, Delft, The Netherlands)
at 10 cm depth. Additional soil moisture and temperature measurements (Trime-Pico 32, Imko, Ettlingen,
Germany) above the heat flux plate at 5 cm depth were used to calculate heat flux at the soil surface. EC
data recording, filtering and processing were carried out identical to the methodology described in Meijide
et al. (2017) for the same study site. As the applied drone-based models all assume full energy balance
closure, we used the Bowen ratio closure method (Pan et al., 2017; Twine et al., 2000) to compute full
closure for the EC measurements.

**2.6 Statistical analyses**

We applied the model II Deming regression method to consider uncertainties in both x and y variables
(Cornbleet and Gochman, 1979; Glaister, 2001) with the assumption that the error ratio ($\sigma\varepsilon^2/\sigma\delta^2$) of the
variances ($\sigma$) of errors on y ($\varepsilon_i$) and on x ($\delta_i$) would not differ from 1 which is the standard procedure if
both uncertainties are unknown (Legendre and Legendre, 2003). We used the interquartile range method
with a factor k=1.5 to remove outliers from the regression. A Durbin-Watson test was applied to test for
correlation in error terms. We checked for heteroscedasticity visually and using a White test. Normal
distribution of error terms was tested visually plotting standardized residuals vs. theoretical quantities and
performing a Shapiro-Wilk test. Standard errors and confidence intervals for slope and intercept of the
Deming regression were calculated using analytical methods (parametric) and the jackknife method
(Armitage et al., 2001; Linnet, 1993). As further indicators of model performance, we calculated the
coefficients of determination ($r^2$), the Mean Absolute Error (MAE), the Root Mean Square Error (RMSE)
and slope and intercept from the Deming regression. Statistics such as $r^2$ have their limitations in method
comparison since they are designed to indicate how well the resulting model of the regression describes
the outcome and are not necessarily a good measure for systematical bias between methods. However,
they are used as a statistic in this study since they represent an additional indicator for interpretation.
Linearity was checked visually plotting residuals vs. fitted values. To examine the spatial heterogeneity
of the LE from drone-data based estimates, we calculated standard deviation, coefficient of variation,
Kurtosis and median-based Fisher-Pearson Coefficient of Skewness (FPCS) for each model output to
characterize the respective distributions in relation to a normal distribution (Doane and Seward, 2011;
Legendre and Legendre, 2003).
All modeling procedures and parts of the statistical analyses were computed using Python version 3.7.1
(Python Software Foundation), involving the libraries NumPy 1.14.2, SciPy 1.1.0, pandas 0.23.1, scikit-
learn 0.19.1, gdal 2.3.2, Astropy 3.2.2 and tkinter 8.6. The Deming regression was computed using the
MethComp and mcr v2.2.1 package (Manuilova et al., 2014) in R version 3.6.1 (R Development Core
Team, 2019). Graphic representation was processed in Python using the Matplotlib 3.0.2 and Seaborn
0.9.0 libraries.





## 2.7 Dataset characteristics

The dataset offers a comparatively high number of replicates from 61 drone recording flights and the corresponding eddy covariance measurements enabling a method comparison which requires at least n = 60 observations (Legendre and Legendre, 2003). The data was recorded in a 30 min frequency, to facilitate the analysis of daily courses of evapotranspiration behavior creating a trade-off situation of more flights per day with shorter flight times per flight. Because flight times were so short, only a smaller footprint with a radius of 100 m around the eddy covariance station was covered, while the footprint recorded with the eddy covariance system ranged up to a 500 m radius around the tower. Therefore, the reduced area of the drone recorded LST maps is often smaller than the extent of the eddy covariance footprint. We have several reasons to assume that this doesn't cause major problems for the comparison though: the study area is very homogenous with an elevation difference of 5 m in the eddy covariance footprint and the biosphere is strongly dominated by only one species (oil palm). The plantation is very well managed, so that all oil palm canopies are vital, no oil palms have died and only dry leaves are removed. A further limitation of the dataset is the lack of morning or night LST measurements that could not be recorded with the drone due to security concerns and limited access to the plantation at night. This doesn't affect the procedure of the DATTUTDUT and TSEB-PT model, but morning measurements are an important factor for the DTD model. We were able to record night and morning measurements with two stationary thermal cameras that were attached to the tower. As for the DTD model, morning and later recordings should ideally be recorded with the same camera. To check whether both cameras would measure similar temperatures, we compared a total of 122 LST maps from the drone and the stationary cameras and plotted the measured and adjusted temperatures of both recording systems (Fig. A1). There is a small deviation of the measured temperatures resulting in a mean absolute error (MAE) and root mean squared error (RMSE) of 1.59 and 2.15 K respectively. Since LST measurements are subject to a certain degree of uncertainty and thermal cameras usually have a measurement error of up to ±1°C we decided to use the morning measurements from the tower cameras as input for the morning temperature reference (Aubrecht et al., 2016). The implementation of the DTD model is therefore strictly experimental and has to be interpreted with the uncertainties of the morning measurements in mind.



## 3 Results

### 3.1 Meteorology

During our 61 flight missions, cloudiness was variable from clear sky to full cloud cover; short-wave irradiance ranged from 204 to 1110 W m$^{-2}$. The prevailing wind direction was from north-east, at an average wind speed of 1.7 m s$^{-1}$. Canopy air temperature ranged from 22.5 to 32.3°C and relative humidity varied between 62 and 99%. The energy balance closure of the reference EC measurements was 0.77 (r$^2$ = 0.87).

### 3.2 Drone-based modeling methods vs. eddy covariance method

At the time of the drone flights, LE from the EC method ranged between 87 and 596 W m$^{-2}$ (mean: 337 W m$^{-2}$). Congruence of the LE estimates of the three applied models differed in their congruence with EC measurements, depending on the configuration of the Rn assessment (Fig. A3). Generally, error metrics were reduced and congruence was increased the more measurement-controlled the Rn determination process was.

For the Rn_mod configuration the daily patterns of DATTUTDUT LE estimates closely agreed with EC measurements around noon but resulted in higher LE fluxes in the morning and afternoon where TSEB-PT LE estimates are much higher than EC measurements (Fig. 2a). While some of the LE predictions from the DTD model in Rn_mod configuration are similar to the EC measurements, many of its LE predictions are over- or underestimations. Models with Rn_sw configuration produced LE estimates that matched LE from EC more closely. DATTUTDUT computed higher estimates of LE compared to the EC measurements during noon, while TSEB-PT produced more congruent LE estimates for the midday hours but rather underestimated LE fluxes especially in the morning. The DTD model underestimated LE fluxes for all daytimes (Fig. 2b). All models with Rn_sw configuration yielded comparably low estimates during the morning and afternoon hours. With Rn_mes configuration, DATTUTDUT computed closely matching LE estimates at all times of day across the five-day measurement period, while TSEB-PT consistently produced much higher estimates than EC around noon and the DTD model underestimated LE fluxes especially in the morning hours while otherwise producing accurate results (Fig. 2c).





**Figure 2:** Latent heat flux (LE) from energy balance models (DATTUTDUT, TSEB-PT, DTD) and three different configurations of net radiation (Rn) determination (Rn_mod, Rn_sw, Rn_mod) and eddy covariance measurements (EC) over five consecutive days (n = 61 flight missions).

Across all daytimes and weather conditions (n=61 flight missions), congruence among drone-based LE





estimates and reference EC measurements was highest for the DATTUTDUT model with Rn_mes
configuration (r²=0.85); MAE and RMSE were 47 and 60 W m$^{-2}$, respectively (Fig. 3). To compare the
model predictions and the eddy covariance measurements, we computed a Deming regression between
both LE predictions and LE estimates. The methods are considered to be statistically interchangeable if
the confidence intervals of the slope and intercept include one and zero respectively. Deming regression
of the LE estimates of the DATTUTDUT model with Rn_mes configuration showed no significant
proportional or constant error compared to EC measurements as the values one and zero lay within the
respective 99% confidence interval ranges of slope and intercept (Fig. 4). It is thus indicated that there is
no significant difference between DATTUTDUT with Rn_mes configuration and the EC technique. The
TSEB-PT model in Rn_mes configuration also showed no significant continuous errors but was subject
to proportional bias (Fig. 4c) predictions tend to overestimate LE around noon when fluxes are very high
(Fig. 2c and 3c). The DTD model showed no proportional bias but indicated a continuous error in the
analytical method and the Jackknife method (Fig. 4c). In the Rn_sw configuration, all three models
showed no significant proportional error of LE estimates compared to EC measurements (Fig. 4b).
However, all TSEB-PT and DTD model estimates deviated from the EC measurements by a significant
constant amount (Fig. 3b and 4b). All models in the Rn_mod configuration showed significant
proportional and constant errors or large biases compared to EC measurements, as well as very large
confidence intervals Fig. 3a and 4a).




**Figure 3:** Model II Deming regression of latent heat flux estimates from drone-based energy balance models (DATTUTDUT, TSEB-PT, DTD) and different configurations of net radiation (Rn_mod, Rn_sw, Rn_mes) with the eddy covariance method (n = 61 flight missions).





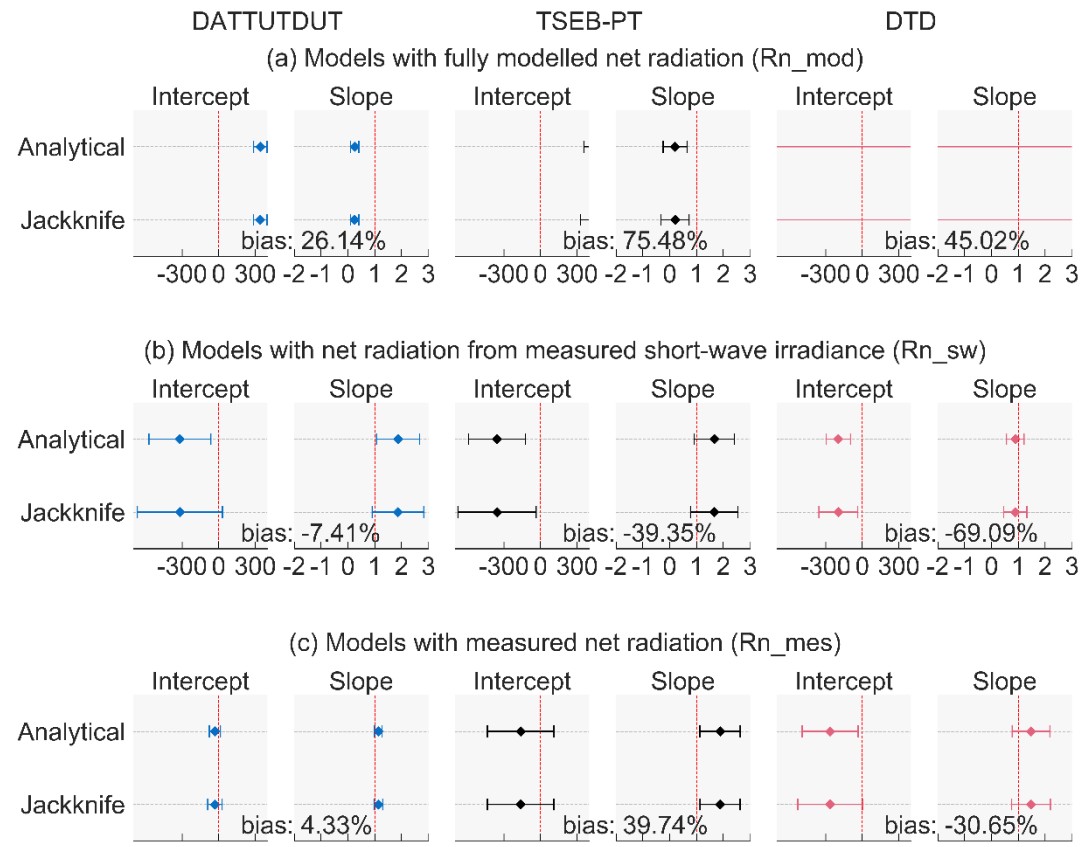

350

**Figure 4:** Confidence intervals for intercept and slope of Deming regression for the different LE
estimation approaches compared with EC measurements. X-level for the bias is the mean of the
established EC reference method. The intercept is displayed in W m$^{-2}$.

354

**3.3 Spatial distribution of LE**

356

For 9$^{th}$ of August 2017, 12.30 h, the DATTUDUT in Rn_mes configuration suggested a mean of 526 W
m$^{-2}$ (minimum of 0 on the corrugated iron roof of the EC tower system, maximum of 637 W m$^{-2}$,
coefficient of variation 7.53 %, for the analyzed 18,383 pixels) (Fig. 5), which translates to a mean ET of
0.778 mm m$^{-2}$ h$^{-1}$. Locally, i.e. in the center of oil palm crowns, high LE of > 400 W m$^{-2}$ was observed,
while LE from soil and ground vegetation areas between oil palm canopies was lower. The LE fluxes of
all pixels were almost normally distributed for the one-source energy balance model DATTUDUT (Fig.
6), whereas the distributions of the two-source energy balance model TSEB-PT (for the same LST dataset)





was more skewed, with more LE observations at the upper end of the range. The DTD model resulted in
an almost normal distribution but with values far more evenly distributed over the entire range from
minimal to maximal value and with a less pronounced peak. While the averaged FPCS ranged around ±1
for all three models and model configurations over the 61 flights indicating only a minor skewness,
Kurtosis was always within the platykurtic distribution for the DATTUTDUT and DTD models revealing
a very low number of outliers whereas the distribution was highly variable for TSEB-PT (Fig. A2).

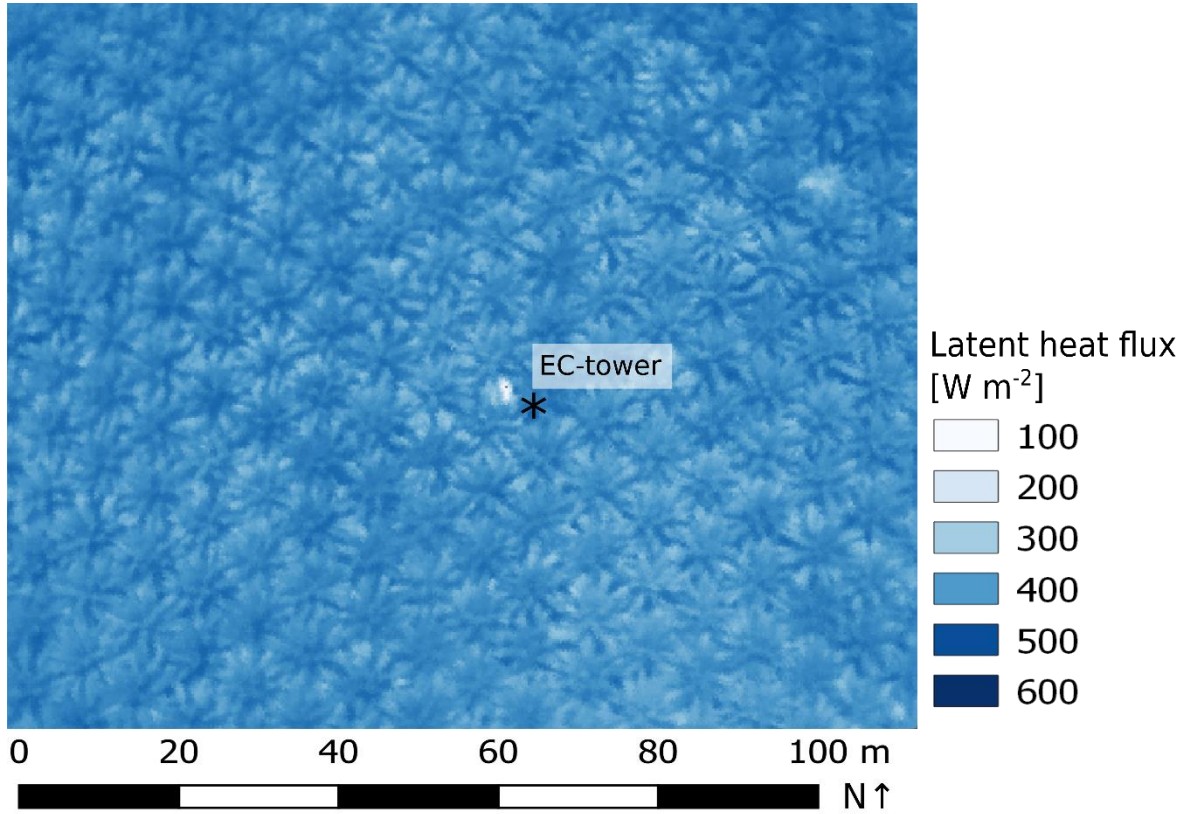


**Figure 5:** Spatial distribution of latent heat flux from drone-based thermography and subsequent energy
balance modeling (DATTUTDUT with Rn_mes configuration, 9 August 2017, 12.30 h).



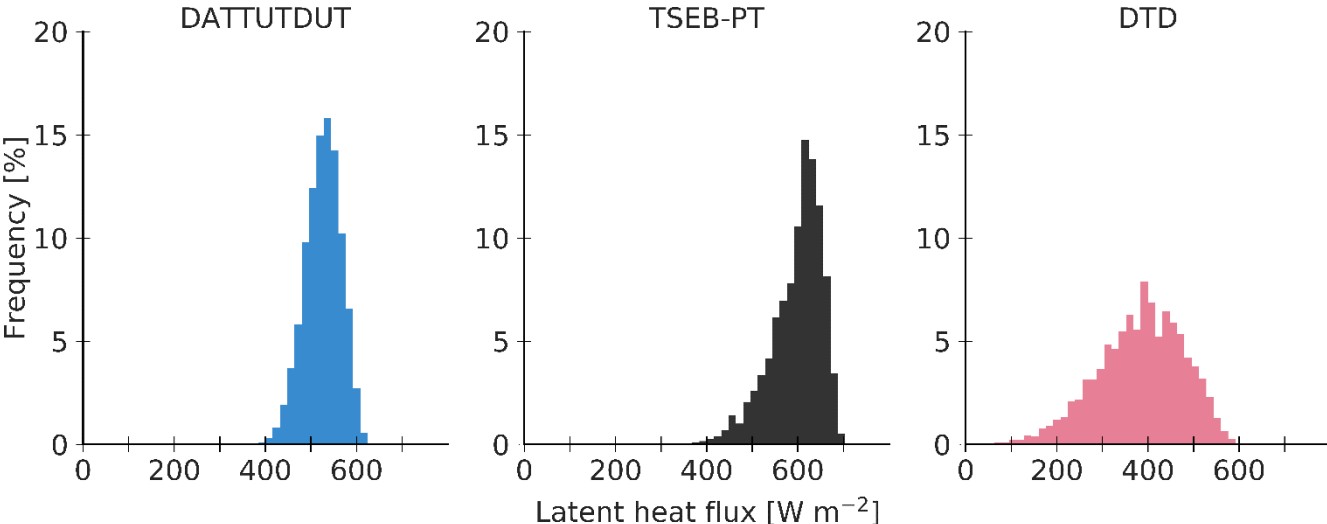

**Figure 6:** Frequency distribution of latent heat flux for the model output images from the same thermal image as shown in Fig. 5 (9 August 2017, 12.30 h). Absolute histogram bin size was set to 16 W m$^{-2}$, we used 50 bins from 0 to 800 W m$^{-2}$.

## 4 Discussion

Our study indicates a high agreement between latent heat fluxes assessed by drone-based thermography and the eddy covariance technique. However, the performance of the three applied energy balance models differed among each other and among different configurations of net radiation assessments in the models (Fig. 2 and A3). Model II Deming regression analyses and associated quality assessments suggest interchangeability between the DATTUTDUT model in Rn_mes configuration and the EC technique (Fig. 3 and 4). Applying this configuration, a fine grain spatial analysis of latent heat fluxes suggests relatively low heterogeneity of LE in the studied tropical oil palm plantation (Fig. 5).

### 4.1 Drone-based LE modeling vs. eddy covariance measurements

The confidence intervals of slope and intercept of the Deming regression indicate that the one-source energy balance model DATTUTDUT with Rn_mes configuration is statistically interchangeable with the established EC method for estimating LE fluxes. There are advantages and limitations to both methods. For example, the DATTUTDUT model provides insights on the spatial distribution of LE fluxes within the full extent of the available LST maps, whereas the EC technique averages the LE fluxes within its footprint to a single value. On the other hand, the DATTUTDUT model is temporally limited to the availability of LST maps, whereas the EC method can measure fluxes continuously over several years





once the equipment is in place. The DATTUTDUT model with Rn_mes configuration further requires
additional measurements of short- and long-wave radiation budgets. In our study, these measurements
were taken with the EC equipment, but future stand-alone drone approaches are possible by using on-
board miniaturized radiation sensors (Castro Aguilar et al., 2015; Suomalainen et al., 2018). The two-
source energy balance models TSEB-PT and DTD in the Rn_mes configuration showed different
behaviors. TSEB-PT was found to have no significant continuous errors, but proportional errors compared
to the reference EC method. This is largely rooted in the overestimation of relatively high fluxes around
noon, while lower fluxes during the morning and afternoon hours were predicted more accurately.
An opposite situation was found for the DTD model that showed no proportional errors but a continuous
error for the analytical method. Intercept confidence intervals from the Jackknife method included zero
and suggest no continuous error for the DTD model in the Rn_mes configuration. Due to the mismatch
between the results of the Jackknife and the analytical method, a considerable bias of -30.65% over the
mean and remaining accuracy concerns of the morning LST measurements, this method configuration
cannot yet be considered as statistically interchangeable.
All models with the Rn_sw configuration showed a significant constant error compared to EC
measurements, i.e. all modeled LE estimates derived from this configuration underestimate measured
fluxes by a certain fixed amount (about 200 W/m² on average compared to EC measurements). These
underestimations of Rn translated directly to an underestimation of turbulent fluxes (Fig. A3). Previous
studies have pointed out that Rn derivation based on short-wave irradiance measurements is challenging
as long-wave radiation budgets are often completely independent from their short-wave counterparts
(Hoffmann et al. 2016). Estimation errors in long-wave radiation budgets have e.g. been reported to be
related to high relative air humidity, when some of the original model assumptions are no longer met
(Hoffmann et al., 2016). We observed a negative correlation (r² = 0.46) between incoming long-wave
irradiance and relative humidity and assume that the high relative humidity in our tropical study area may
have affected the determination of Rn when using the Rn_sw configuration through inaccuracies in
estimating long-wave radiation budgets, therefore causing the observed significant continuous errors.
Such constant errors in Rn estimation can be reduced or even eliminated by enhanced calibration of the
models. We thus also consider the Rn_sw configuration valuable for future research, particularly because
measurements of incoming short-wave radiation are much easier to implement than assessing complete
short- and long-wave radiation budgets as necessary for the Rn_mes configuration. The application of the
Rn_sw configuration for a one-source energy balance model such as DATTUTDUT was also tested in
two previous studies, with similar results to our study, i.e. a reduction of errors compared to its original
formulation with fully modeled Rn_mod (Brenner et al., 2018; Xia et al., 2016).
Lastly, the model configuration Rn_mod did not produce accurate LE estimates for all three models, as
many of the basic assumptions for fully modelled Rn determination are not met in tropical environments
such as our equatorial study area. As such, the sky is often cloudy, while haze frequently occurs during
periods without rainfall. Even if no clouds are visible, relative humidity is often high, which interferes



with the clear-sky assumptions of the Rn_mod configuration (Still et al., 2019).

LE estimates from TSEB-PT and DTD are sometimes close to zero or even negative (for all three Rn configurations) and thus deviate substantially from the EC measurements (Fig. 2). This occurs when the difference between surface aerodynamic temperature and air temperature becomes very small, which causes the evaporative fraction to approach zero, resulting in an overestimation of H and an underestimation of LE in the fragmentation process of turbulent fluxes (see Eq. 1 in Norman et al. 2000). This effect is especially pronounced for the DTD model, when the morning reference air and land surface temperature measurements are very similar to each other. We assume that the problem is rather typical for tropical environments as daily temperature changes are often not as pronounced as e.g. in semiarid areas where, conversely to our findings substantial overestimations of LE fluxes were observed (Kustas et al., 2016; Morillas et al., 2013).

Among the three models applied in our study, the relatively simple DATTUTDUT model produced the most precise LE estimates compared to eddy covariance reference measurements. Similar conclusions were reached by Brenner et al. (2018), where DATTUTDUT marginally outperformed the more complex TSEB-PT model. On the other hand, contrasting observations were made by Xia et al. (2016) in vineyards with more extreme temperature divergences between soil and vegetation, where the TSEB-PT model produced more precise estimates of LE than the DATTUTDUT model. This was explained by the better physical representation of energy and radiative exchange in the TSEB-PT model. The authors further point out that Rn determination is not the only source of error in the DATTUTDUT model (Xia et al., 2016). In our study, the TSEB-PT model slightly outperformed the more complex DTD model in the Rn_mes configuration regarding error terms, congruence and continuous errors. The DTD model on the other hand showed less bias and no proportional error and is therefore the more promising approach as its continuous error can be reduced through calibration.

We used the Bowen-ratio method to close the energy balance for the reference EC measurements. As reported by Xia et al. (2016), agreement between measured EC and modeled LE estimates could potentially be increased by using the residual method from Twine et al. (2000) for energy balance closure. Further potential improvements include the aerial sampling alignment with the EC measurement logging cycles. We compared snapshot measurements of LST to 30 min averages of EC for the corresponding times in an environment where key variables such as solar irradiance can change very quickly. Better matching the measurement cycle duration may further improve agreement between the methods and was already suggested in a previous study (Brenner et al., 2018). Further, in our study the aerial-derived LST images represented only the center area of the (at times quite variable and large) EC footprint. Covering the whole potential area of the footprint in all directions could also increase agreement between the measurements, but would require even higher flight altitude or longer flight times to cover the whole area; both options would reduce the number of temporal replicates and increase errors from measurements and





processing, but could nonetheless be viable approaches for other research questions.

Only few previous studies have demonstrated applicability and limitations of estimating LE with the three
energy balance models from non-satellite data. In these studies, LSTs were e.g. recorded from drones for
European grasslands and croplands (Brenner et al., 2018; Hoffmann et al., 2016) and from drones or
airplanes for taller vegetation including olive orchards and vineyards (Ortega-Farías et al., 2016; Xia et
al., 2016). Our study adds to this an application of these models in a tropical environment, for higher
vegetation (i.e. oil palm) and across variable daytimes and weather conditions. We further analyzed for
the first time whether drone-data based models and EC measurements can be used interchangeably, as
our large sample size of n=61 flights allowed for a method comparison based on a model II Deming
regression (Legendre and Legendre, 2003). We conclude that this is the case for some models and
configurations, above all for the DATTUTDUT with Rn_mes configuration.

**4.2 Spatial distribution of latent heat fluxes**

A particular strength of drone-based thermal imagery is the high spatial resolution which allows for
spatially explicit assessments of evapotranspiration, within and potentially also beyond the footprints of
EC towers. The outlines of the single oil palm canopies are clearly visible in the LE flux map (Fig. 5),
with the highest LE fluxes occurring in the center of the oil palm canopies. We assume that this spatial
pattern is caused by an increased local LAI in the centers of the oil palm canopies, while leaf area density
decreases towards the outer canopies. Further, the central areas of oil palm canopies are more exposed to
sunlight and wind throughout most of the day, increasing their potential for (evapo)transpiration
compared to canopy edges. Mixed pixel effects (among soil and canopy) likely also contribute to the
observed lower LE fluxes towards the borders of oil palm canopies. Further contributing factors to higher
LE fluxes in the centers of oil palm canopies could be leaf age (with younger leaves in the center) and
additional ET from pockets in the axils of pruned leaves along the stem, which contain small water
reservoirs and epiphytes (Meijide et al., 2017; Tarigan et al., 2018).

In the histograms of LE fluxes from all pixels within the single studied footprints (Fig. 6), the
DATTUTDUT and DTD models results in a bell-shaped normal distribution but very different value
ranges. While the DATTUTDUT histogram shows only few pixel values of zero and most pixels closely
distributed around the mean, the DTD histogram is much wider distributed and the peak is much more
moderate. Mean and median are very similar indicating close to zero skewness for both, the
DATTUTDUT and DTD model. Such a distribution tending towards unimodality is also considered
typical for landscapes where ET is highly dominated by one species (Xia et al., 2016). The TSEB-PT
model shows a different, more skewed distribution of LE fluxes (for the same dataset of LST), with the
median of the LE estimates located between the mean and the upper end of the LE flux range. Kurtosis is





much more pronounced for the TSEB-PT model in all net radiation configurations than in the other two
models (Fig. A2).
The increased Kurtosis is a very strong sign towards more extreme outliers that emerge in the TSEB-PT
model. We assume that the TSEB-PT model is more sensitive to dry surfaces and hence produces more
extreme outliers. Since the DTD model references the LST measurements with a second set of land
surface and air temperatures it is less affected by extreme outliers in the LST data.

We see great potential in the drone-based remote sensing applications presented in this study; especially
when recent developments in drone-environment interaction, mobile edge computing (potentially on-
board of the drone) and communication technologies such as LoRaWan (Long Range Wide Area
Network) or 5G are combined (Becerra, 2019; Marchese et al., 2019). Autonomous acquisition of LSTs
over EC stations and the surrounding areas can be supplemented by on-board and ground sensors and
energy-balance models can be computed on the edge enabling a dense temporal resolution of LST, flux
and ET maps in almost real-time. This concept can e.g. be used for the attribution of fluxes in mixed
species plant communities, the study of edge effects in landscapes, and further be adapted e.g. to detect
water stress in agriculture and forests.

**5 Conclusions**

Drone-based thermography and subsequent energy balance modeling under certain configurations can be
considered a highly reliable method for estimating latent heat flux and evapotranspiration; for some
configurations statistical interchangeability is suggested with the established eddy covariance technique.
They thus complement the asset of available methods for evapotranspiration studies by fine grain and
spatially explicit assessments.






## Appendix I

**Table A1:** All flight missions (n = 61) indicated with their flight registration numbers (176-236) conducted from DOY 217 to DOY 221 in 2017. (c) indicates cloudy or partly cloudy skies as observed on site.

Timetable PTPN VI Flights (176-236)

| local time | 217 | 218 | 219 | 220 | 221 |
|---|---|---|---|---|---|
| 9:00 | - | (c) 189 | (c) 200 | (c) 213 | (c) 223 |
| 9:30 | - | (c) 190 | (c) 201 | (c) 214 | (c) 224 |
| 10:00 | 176 | (c) 191 | (c) 202 | (c) 215 | (c) 225 |
| 10:30 | 177 | 192 | 203 | (c) 216 | (c) 226 |
| 11:00 | 178 | 193 | (c) 204 | (c) 217 | 227 |
| 11:30 | (c) 179 | 194 | 205 | 218 | 228 |
| 12:00 | (c) 180 | 195 | 206 | 219 | (c) 229 |
| 12:30 | 181 | (c) 196 | 207 | 220 | 230 |
| 13:00 | 182 | 197 | 208 | 221 | 231 |
| 13:30 | 183 | 198 | (c) 209 | (c) 222 | (c) 232 |
| 14:00 | 184 | 199 | (c) 210 | - | (c) 233 |
| 14:30 | - | - | (c) 211 | - | (c) 234 |
| 15:00 | (c) 185 | - | (c) 212 | - | (c) 235 |
| 15:30 | (c) 186 | - | - | - | (c) 236 |
| 16:00 | (c) 187 | - | - | - | - |
| 16:30 | (c) 188 | - | - | - | - |





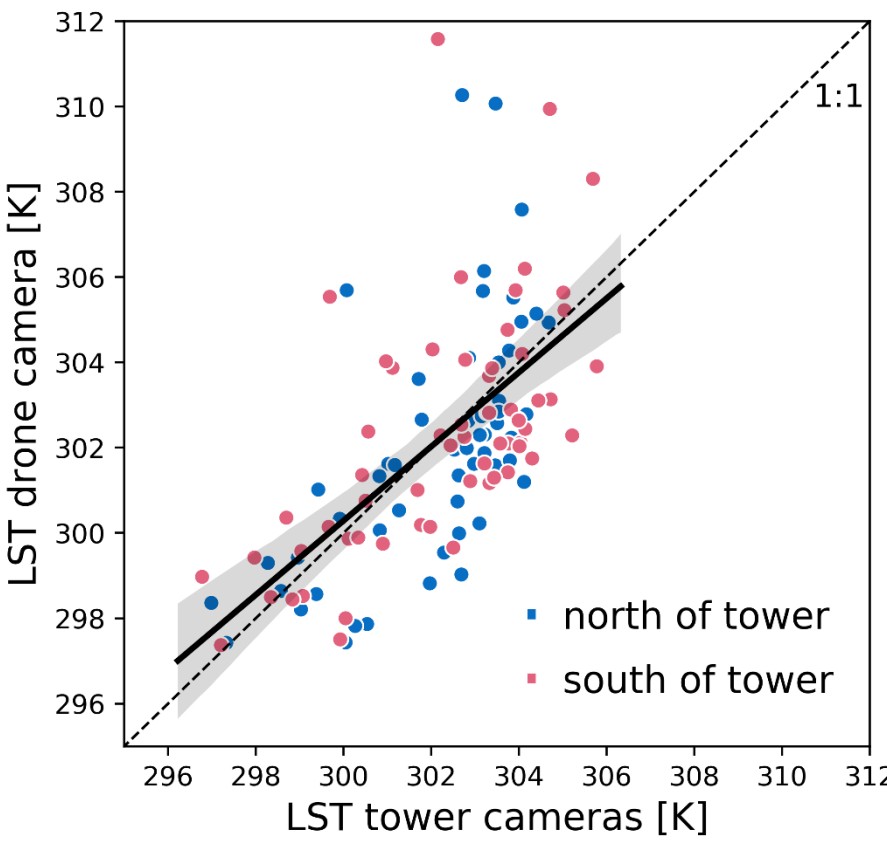

541

**Figure A1:** Scatterplot and linear regression of LSTs obtained with the drone and tower cameras. Mean
absolute error (MAE) is 1.59 K and root mean squared error (RMSE) is 2.15 K, relationship of both
measurements is $r^2 = 0.4$.







**Figure A2:** Boxplot distribution of histogram Kurtosis and Fisher-Pearson Coefficient of Skewness (FPCS) for the different model settings, each with n=61. Zero for normal distribution is indicated with a red line, the blue area indicates platykurtic distribution.





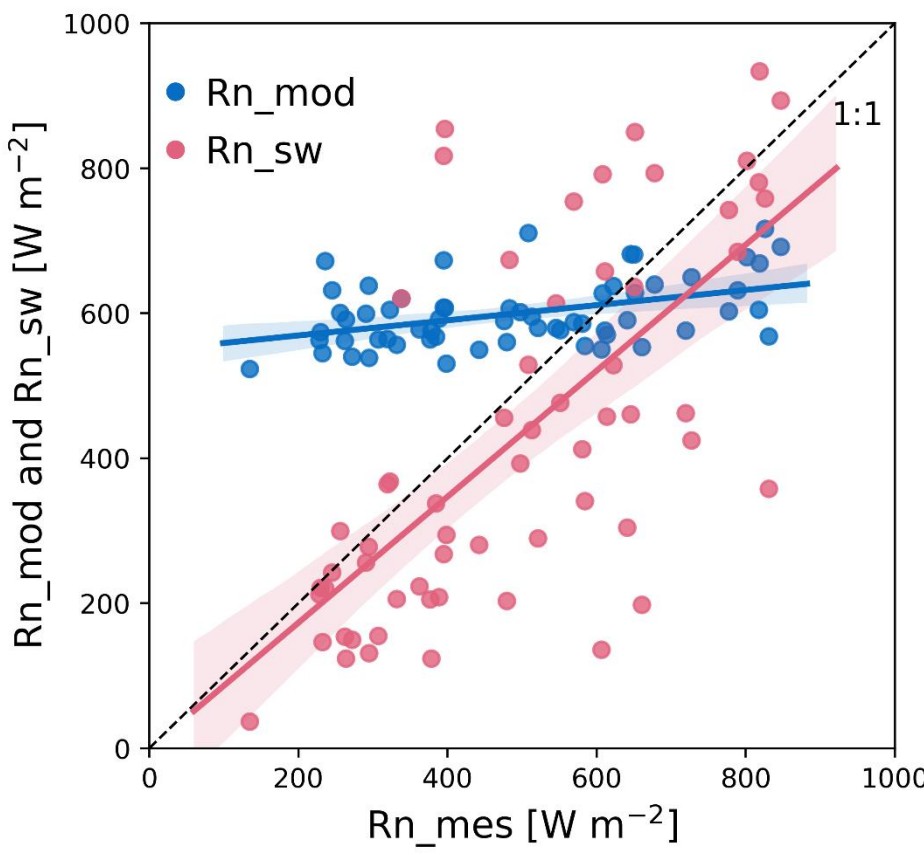

550

**Figure A3:** Measured net radiation (Rn_mes) plotted against fully modeled net radiation (Rn_mod) and

net radiation estimates based on short-short wave irradiance (Rn_sw).

553





## Appendix II

## Method appendix: model descriptions

All models in this study use instantaneous land surface temperatures (LST) to solve the energy balance equation:

$R_n = G + H + LE$ (eq. 1)

Where $R_n$ is the net radiation, G is the ground heat flux and the turbulent fluxes H and LE represent sensible and latent heat flux, respectively. $R_n$ is estimated by calculating the budget of incoming (↓) and outgoing (↑) long- (l) and short-wave (s) radiation:

$R_n = R_s{\downarrow} + R_s{\uparrow} + R_l{\downarrow} + R_l{\uparrow} = (1- \alpha) * R_s{\downarrow} + \varepsilon_{surf} * \varepsilon_{atm} * \sigma * T_{air}{}^4 - \varepsilon_{surf} * \sigma * T(\theta)_{surf}{}^4$ (eq. 2)

where the short-wave component is calculated by multiplying incoming shortwave radiation $R_s{\downarrow}$ [W m$^{-2}$] with its absorption ratio deducted from the combined soil and vegetation albedo α. The long-wave radiation budget is calculated from surface (soil and vegetation) emissivity $\varepsilon_{surf}$ and atmospheric emissivity $\varepsilon_{atm}$, the Stefan-Boltzmann constant σ [$5.6704*10^{-8}$ W m$^{-2}$*K$^{-4}$], air temperature $T_{air}$ and radiometric land surface temperature $T(\theta)_{surf}$ (both in K). The available energy consisting of the turbulent fluxes H and LE is calculated by subtracting G from Rn. G is computed as a linear function of Rn similar as described in Liebethal and Foken, (2007):

$G = a * R_n – b$ (eq. 3)

We set a = 0.1 for ground heat flux under canopies and b = 0 (Ogée et al., 2001). A time offset $\Delta R_n$ as in the original formulation of Liebethal and Foken, (2007) is not included for simplicity reasons. With the fraction of turbulent fluxes ($R_n$ – G / $R_n$) known, radiometric LSTs are used to calculate H and obtain LE as a residual.

2.3.3 DATTUTDUT

Key input for the DATTUTDUT model is a LST map from where the hottest and the 0.5% of coldest pixels are extracted, assuming that hot pixels are a result of very little to no evapotranspiration and cold pixels origin in a high evapotranspiration rate (Timmermans et al., 2015). Fully modeled $R_n$ is calculated based on down-welling short-wave radiation estimates calculated using sun-earth geometry to solve eq. 2. Surface albedo $P_0$ is calculated as in Timmermans et al. (2015) based on the assumption that dense



vegetation appears colder than rocks or soil in the thermal imagery (Brutsaert, 1982; Garratt, 1992):

$P_0 = 0.05 + ((T_0 - T_{min}) / (T_{max} - T_{min})) * 0.2$       (eq. 4)


Down-welling shortwave radiation $R_s{\downarrow}$ is calculated from the dimensionless atmospheric transmissivity
$\tau$ and the exo-atmospheric shortwave radiation $SW_{exo} = 1360$ W m$^{-2}$ (Timmermans et al., 2015).
Transmissivity $\tau$ is calculated as described in Burridge and Gadd (1977) using the solar elevation angle $\alpha$
that was determined from the geographic position of our site and the coordinated universal time (UTC)
of the measurements:

$\tau = 0.6 + 0.2 * \sin(\alpha)$  (eq.5)

$R_s{\downarrow} = \tau * SW_{exo}$      (eq. 6)

Timmermans et al. (2015) suggest using a constant value of 0.7 for $\tau$ and 0.8 atmospheric emissivity
($E_{atm}$), but as our flight times range from 09:00 to 16:30 h local time we decided to include the solar
elevation angle as in eq. 5. Further, we used a constant surface emissivity ($E_{surf}$) of 0.98 and not 1.0 as in
the original formulation of the DATTUTDUT model. Air temperature $T_{air}$ was calculated as mean
temperature from 0.5% of the coldest pixels in the image.

Calculation of $R_n$ using short wave irradiance or measured $R_n$ for DATTUTDUT:

As the original DATTUTDUT formulation doesn't account for cloud cover, eq. 6 is replaced by measured
short-wave irradiance as in Brenner et al. (2018) for model runs with Rn_sw. For model runs with Rn_mes
eq. 2 was replaced by Rn measurements recorded at the EC-tower.

Calculation of turbulent fluxes in DATTUTDUT:

The sum of the turbulent fluxes is calculated by subtracting G from Rn. The result is fractioned into its
components H and LE, using the evaporative fraction (EF) (Timmermans et al., 2015):

$EF = LE / (LE+H) = LE / (Rn - G) = (T_{max} - T(\theta)_{surf}) / (T_{max} - T_{min})$       (eq. 6)

TSEB-PT

The TSEB-PT model divides eq. 1 into a canopy and a soil fraction (Kustas and Norman, 1999; Song et





al., 2016; Xia et al., 2016). The model consists of two parts: First an initialization part where all
parameters that do not depend on soil and canopy temperature partition and knowledge of atmospheric
stability are computed. Afterwards an iterative part where the Monin-Obukhov length is stabilized and
the fluxes are finally derived. To begin this process vegetation cover $f_c(\theta)$ is computed as in Campbell
and Norman, (1998):
$f_c(\theta) = 1 - \exp((-0.5\Omega(\theta) * LAI) / (\cos(\theta)))$ (eq. 7)
where LAI is leaf area index, $\theta$ is the sun zenith angle and $\Omega$ is a nadir view clumping factor to represent
the cross-row structure in which the oil palm is planted (Kustas and Norman, 1999). Guzinski et al.,
(2014) suggest a maximum limit of 0.95 for $f_c(\theta)$, so that a small fraction of the soil is still visible and
extreme magnitudes for soil temperature are avoided. Roughness parameters are calculated from
vegetation height. Eq. 2 is used for the original calculation of $R_n$ from short-wave irradiance for TSEB-
PT. $T_{air}$ was measured at the EC-tower, $T(\theta)_{surf}$ was recorded with the drone both similar to descriptions
in Hoffmann et al. (2016). The canopy emissivity $E_{leaf}$ was set to 0.98 and soil emissivity $E_{soil}$ to 0.95.
The three resistances in the soil-canopy-atmosphere heat flux network, the aerodynamic resistance to heat
transport ($R_A$), the resistance to heat transport from the soil surface ($R_S$) and the total boundary layer
resistance of the leaf canopy ($R_X$) are calculated as in Norman et al. (2000, 1995). Net radiation and the
three resistances remain constant during the model runs. After finishing the computation of all constant
parameters, the iterative part of the model starts assuming Monin-Obukhov length tends to infinity. In the
first iteration $R_n$ is partitioned into a soil and canopy fraction by calculating net radiation divergence $\Delta R_n$
(Hoffmann et al., 2016; Norman et al., 2000):
$\Delta R_n = R_n * (1 - \exp((-K * LAI * \Omega 0) / \sqrt{(2\cos(\theta_s))}))$ (eq. 8)
where K is an extinction coefficient that varies according to LAI (Hoffmann et al., 2016). We are aware
of the fact, that the determination of K using LAI is disputed as other studies found no significant
correlation of K and LAI (Zhang et al., 2014). With $\Delta R_n$ known, sensible heat flux is then estimated using
the Priestley-Taylor approximation:
$H_c = \Delta R_n * (1 - \alpha_{PT} * f_G * (D/(D+\gamma))$ (eq. 9)
Where $\alpha_{PT}$ is the Priestley-Taylor coefficient that was calculated as described in Lhomme (1997). The
psychrometric constant $\gamma$ and the slope of the saturation pressure curve D were both calculated as in Allen
et al. (1998). Canopy temperature $T_C$ was computed by summing up the results of the linear approximation
in equation (A7) for $T_{C,lin}$ and $\Delta T_C$ from equation (A11) both from Norman et al. (1995). Knowing canopy
temperature $T_C$ and fraction of view covered by vegetation $f_\theta$ as in Hoffmann et al. (2016), soil





temperature $T_S$ can be calculated:

$T_s = (T(\theta)_R{}^4 - f_\theta * T_C{}^4) / (1-f_\theta)^{(1/4)}$                    (eq. 10)

Now with soil and canopy temperatures and the resistances of the soil-canopy-atmosphere heat flux
network known, fluxes can be calculated with equations (9), (10), (11) and (13) from Hoffmann et al.
(2016). Soil heat flux G is computed as in equation (3). Total latent and sensible heat fluxes are calculated
as the sums of canopy and soil fluxes. In the following iterations, a recalculation of Monin-Obukhov
length takes place until a stable value is reached and the resulting fluxes are derived.



Turbulent fluxes are calculated and the original EF is also used to estimate LE from fully modeled and
measured $R_n$. For model runs with fully modelled $R_n$, $R_n$ is calculated as in DATTUTDUT using equation
(2), for model runs with measured Rn, Rn is measured at the EC-tower.


DTD

The Dual-Temperature-Difference (DTD) model works very similar to TSEB-PT and differs mainly in
the way how sensible heat flux is calculated (Hoffmann et al., 2016). A detailed description of the model
can be found in Guzinski et al. (2014) and Norman et al. (2000). It uses two observations of LST and air
temperature in the same day. The first observation is recorded in the early morning hours, when fluxes
are known to be minimal and the second observation is recorded later on the same day at any given time.
To account for inaccuracies and bias in the measurements a double-difference of LSTs and air
temperatures is calculated to avoid the direct use of absolute LST data. Equation (A37) from Guzinski et
al. (2014) describes how sensible heat flux is calculated using a series resistance network instead of the
parallel from the original model formulation in Norman et al. (1995).













Calculation of evapotranspired amount of water:

The actual amount of evapotranspired water ($ET_w$) in mm h$^{-1}$ was calculated as in Timmermans et al. (2015):

$$ET_w = ((LE*t)/1000000)/(2.501-0.002361*(T_{air}-273.15))$$

Where LE is the latent heat flux in W m$^{-2}$, t is the respective timespan in seconds and $T_{air}$ is the air temperature in Celsius.

**Data availability**

The final data used for the statistical tests were uploaded in Göttingen Research Online Data with a doi: https://doi.org/10.25625/JY9AFT
Raw thermal images, orthomosaics and terrain data, georeferenced rasters and model configurations are available upon request to the corresponding author.

**Author Contribution**

The study was conceptualized by DH in cooperation with H (drone measurements) and AK in cooperation with TJ (eddy covariance measurements). FE led the writing of the paper with help from AR and DH supervised the work. FE collected and processed the drone data and CS the eddy covariance data. FE conducted data processing, model application, statistical analysis and production of plots in cooperation mainly with DH and AR. FE, DH and AR created a first version of the manuscript, which was further improved in a cooperation of all authors.

**Competing interests**

The authors declare that they have no conflict of interest.





## Acknowledgements

This study was funded by the Deutsche Forschungsgemeinschaft (DFG, German Research Foundation) – project number 192626868 – SFB 990 (subprojects A02 and A03) and the Ministry of Research, Technology and Higher Education (Ristekdikti). We thank Ristekdikti for providing the research permit for field work (No. 322/SIP/FRP/E5/Dit.KI/IX/2016, No. 329/SIP/FRP/E5/Dit.KI/IX/2016 and No. 28/EXT/SIP/FRP/E5/Dit.KI/VII/2017). We thank our field assistants Zulfi Kamal, Basri, Bayu and Darwis for great support during the field campaigns and Edgar Tunsch, Malte Puhan, Frank Tiedemann and Dietmar Fellert for their technical support. We thank Hector Nieto for publishing the code for TSEB-PT and DTD (pyTSEB) on www.github.com. We also thank Perseroan Terbatas Perkebunan Nusantara VI, Batang Hari Unit (PTPN6) for giving us permission to conduct our research at the oil palm plantation. Thanks to all 'EFForTS' colleagues and friends in Indonesia, Germany, and around the world.





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
