# Peer review of "Predicting evapotranspiration from drone-based thermography – a method comparison in a tropical oil palm plantation"

_Biogeosciences, 2020_

## Referee Comment (RC1) · Tamir Klein (Referee) · 2 Jun 2020

Peer review for the manuscript: Predicting evapotranspiration from drone-based thermography – a method comparison in a tropical oil palm plantation by Ellsasser et al

The manuscript under consideration reports a 9-days study of surface temperature measurements over an oil palm plantation in Indonesia using a thermal camera mounted on a drone. The authors used the temperature data to calculate the latent heat flux using three different models, with/out radiation inputs, and showed good agreement between one of the models and the latent heat estimated from an eddy-covariance (EC) calculation based on an on-site flux tower. The drone-based temperature calculation is more flexible than the EC, also providing spatial information at high resolution.

This is a very nice paper reporting an elegant study. The text and figures are carefully prepared and nicely presented. I have only a few questions and suggestions:

1. Considering the rather narrow variation in air temperature over the tropical plantation, would you think that the fact that the study was performed at this site is a challenge? Or rather an easier case? I think that this point is touched upon, but further discussion would be appreciated.

2. Considering the aggravating situation of deforestation in the studied region, and the implications on surface warming (L110-113), it would be highly interesting to make a comparison study between the palm plantation and the natural rainforest. I assume that the higher spatial heterogeneity in the latter would offer a better test case for the spatial distribution of ET (Fig. 5). Can the authors include such information?

3. It would be good to include in the paper some information on the measured air-surface temperature differences as function of time and space.

4. With 90% canopy cover, LST is mostly that of the leaf surfaces, i.e. reflecting the process of evaporative cooling of leaves by transpiration. Can the authors report these (evapo)transpiration values? A value is given in L360. Why are the units mm h-1 m-2? I thought that the mm already includes the area consideration (i.e., 1 mm = 1 L m-2).

5. By using the EC data as absolute reference, the text seems to assume that the EC data are independently true. However, the EC is also an estimate based on an indirect measurement. If there are any additional measurements that could further constrain these data, it would be very helpful. Regardless, the text should be adjusted to reflect that two estimates are compared, rather than an estimate to a direct measurement.

6. In case that one doesn't have radiation measurements, would the DTD model be the best option to make use of the thermal information? In L400 the authors should note

that such sensors must be tested independently in a separate study.

7. The authors discuss measurements in drier sites. It would be interesting to compare these results with measurements of palm water-use and its effect on temperature. Below are a few studies on date palm, evidencing the high transpiration rates in a plantation, and the effect on temperature in an urban context.

8. Finally, another potential comparison could be made with a study of transpiration of forest trees estimated by spatial temperature data from a thermal camera (see reference below).

Sperling, O., Shapira, O., Cohen, S., Tripler, E., Schwartz, A., & Lazarovitch, N. (2012). Estimating sap flux densities in date palm trees using the heat dissipation method and weighing lysimeters. Tree Physiology, 32(9), 1171-1178.

Potchter, O., Goldman, D., Kadish, D., & Iluz, D. (2008). The oasis effect in an extremely hot and arid climate: The case of southern Israel. Journal of Arid Environments, 72(9), 1721-1733.

Potchter, O., Goldman, D., Iluz, D., & Kadish, D. (2012). The climatic effect of a man-made oasis during winter season in a hyper arid zone: The case of Southern Israel. Journal of arid environments, 87, 231-242.

Lapidot, O., Ignat, T., Rud, R., Rog, I., Alchanatis, V., & Klein, T. (2019). Use of thermal imaging to detect evaporative cooling in coniferous and broadleaved tree species of the Mediterranean maquis. Agricultural and forest meteorology, 271, 285-294.

---

## Referee Comment (RC2) · Anonymous Referee #2 · 8 Jul 2020

General comments

The manuscript by Ellsasser et al. makes an interesting and useful contribution to the burgeoning literature on using UAVs to measure ecosystem properties and processes, in this case measurements of surface temperature for use in models of the surface energy balance to predict spatial variations in the latent heat flux and for comparison to eddy covariance-derived estimates of the same.

The appendix describing the various energy balance/ET models should be better integrated with the main body of the manuscript, and as noted below some of the model equations need more clarification. In general, a reader should not have to read other

previous papers to understand the approaches tested here (e.g., see my comments below regarding lines 174-175).

I agree with the other reviewer that more discussion of the various uncertainties in EC-derived ET need to be discussed. While it is the reference method here it is also subject to many uncertainties.

The writing is generally fine but there are a few very awkward sentences that I suggest re-writing (see below).

Specific comments Lines 90-91: "the hottest and a group of coldest pixels in the image" – This is not and independent clause as it is missing a verb

Lines 105-107: This sentence is confusing and needs to be re-written.

Line 110: replace "presented" with "current"

Line 147: Quote the manufacturer's measurement uncertainty here, as you also discuss it later when mentioning thermal cameras. The true uncertainty is surely closer to 1-2 K for cameras like this.

Line 164: Provide the assumed surface emissivities used in each model and component

Lines 174-175: Need to better explain this approach. P-T is usually used to predict LH fluxes not SH fluxes.

Lines 196-207: Do these models assume a closed energy balance? If so how does that affect your estimates?

Line 219: Was this an aspirated measurement of Tair?

Line 222: These are IRTs not thermal cameras, so you do not know exactly which canopy elements you are measuring! Were they capturing only leaves all of the time? Also, what surface emissivity was assumed for these measurements of surface temperature? Did you correct for the influences of reflected longwave radiation, relative humidity, distance to object, etc? And what are measurement uncertainties of the IRTs?

Line 229: Describe the Bowen ratio closure method in more detail.

Line 247: "systematic"

Line 273: I think you mean "alive"

Lines 278-286: As noted above these measurements were not made with thermal camera but with IRTs. Please update.

Line 280: Is the 122 number based on 2 maps/flight?

Line 293-294: Is this peak SW measured during the flight or average SW?

Line 295: By "canopy air temperature" do you mean the Tair measured at 22m?

Line 302-303: This is an awkward sentence – rewrite.

Line 303: The first time you cite Fig. A3 you need to discuss why modeled Rnet is so poor.

Line 304: Replace "congruence" with "agreement" or "fidelity"

Line 307-308: Perhaps this poor agreement in morning and late afternoon is not surprising since the dATTUDUT method is based on modeled Rnet..?

Line 308-309: It's worth breaking out the description of the performance of the TSEB-PT estimates into a separate sentence. Are these estimates uniformly higher than the EC estimates or only during part of the day?

Lines 335-336: Seems like this sentence is missing a word or two.

Line 352: I'm unclear what you mean about the X-level for the bias in EC reference fluxes.

Lines 405-406: Are you referring to the slope in this sentence?

Lines 455-457: Well before this discussion of errors you should define what you mean by proportional versus continuous errors.

Line 500: Replace "results in" with "predict"

Line 503: eliminate comma after "both"

Line 520: Which edge? Computer or edge of study area?

Line 542: Replace "cameras" with "IRTs"

Line 565: How are the surface epsilon (emissivity) terms estimated? Do they vary spatially across the image?

Lines 579-580: Show the equations for calculating radiometric LSTs.

Line 588: I assume this (Po) is a shortwave albedo?

Line 600: This model assumes cloud-free conditions (with a constant transmissivity)?

Line 605: Is that supposed to be an epsilon symbol as in equation 2?

---

## Author Comment (AC1) · 13 Aug 2020

Peer review for the manuscript: Predicting evapotranspiration from drone-based thermography – a method comparison in a tropical oil palm plantation by Ellsasser et al The manuscript under consideration reports a 9-days study of surface temperature measurements over an oil palm plantation in Indonesia using a thermal camera mounted on a drone. The authors used the temperature data to calculate the latent heat flux using three different models, with/out radiation inputs, and showed good agreement between one of the models and the latent heat estimated from an eddy-covariance (EC) calculation based on an on-site flux tower. The drone-based

temperature calculation is more flexible than the EC, also providing spatial information at high resolution. This is a very nice paper reporting an elegant study. The text and figures are carefully prepared and nicely presented.

Dear Reviewer,

Thank you for taking the time to revise our manuscript. We welcome your comments and think they have helped to improve our manuscript considerably. Please find our point-by point replies below.

Sincerely,
Florian Ellsäßer

I have only a few questions and suggestions:

1. Considering the rather narrow variation in air temperature over the tropical plantation, would you think that the fact that the study was performed at this site is a challenge? Or rather an easier case? I think that this point is touched upon, but further discussion would be appreciated.

We agree with the reviewer that there was rather narrow variation during the time of study (canopy air temperature ranged from 22.5 to 32.3 °C), as is typical for the region. Generally, the study site was rather challenging due to high temperatures and humidity and frequent occurrence of haze, as well as for logistical reasons. Additionally, many previous drone-based studies were conducted on grasslands (e.g. Brenner et al., 2018, 2017) or on low-growing crops such as wheat fields (Hoffmann et al., 2016),
but not on crops with a rather complex canopy structure such as oil palm. However, our study site showed large temperature differences between soil and canopy, which simplified the distinguishing of each fraction.

As suggested, we will add a short section taking up these points to the discussion:

*Generally, the study site was rather challenging due to high temperatures and humidity and frequent occurrence of haze, as well as for logistical reasons. Additionally, many previous drone-based studies were conducted on grasslands (e.g. Brenner et al., 2018, 2017) or on low-growing crops such as wheat fields (Hoffmann et al., 2016), but not on crops with a rather complex canopy structure such as oil palm. On the other hand, our study site showed large temperature differences between soil and canopy, which simplified the distinguishing of each fraction.*

2. Considering the aggravating situation of deforestation in the studied region, and the implications on surface warming (L110-113), it would be highly interesting to make a comparison study between the palm plantation and the natural rainforest. I assume that the higher spatial heterogeneity in the latter would offer a better test case for the spatial distribution of ET (Fig. 5). Can the authors include such information?

We thank the reviewer for this very interesting point. Indeed the comparison of land surface temperatures and modelled evapotranspiration of natural rainforest and an oil palm plantations would provide valuable spatial insight into the current transformation of transpiration patterns caused by local- and regional-scale land-use changes, as e.g. described in Röll et al., 2019 and Sabajo et al., 2017. However, the present study focuses on the comparison of different drone-based methods as a baseline for future ecological studies, rather than applying the methods to different land-use

types. We will however follow up on this in the future, as we also performed flight missions over flooded and non-flooded natural forest sites and a variety of adjacent areas including mixed oil palm stands and small holder rubber and oil palm plantations.

To clarify this point in the manuscript, we will update the introduction section with the following sentence:

*The present study focuses on the comparison of different drone-based methods as a baseline for future ecological studies, rather than applying the methods to different land-use types.*

3. It would be good to include in the paper some information on the measured air-surface temperature differences as function of time and space.

The differences of mean land surface and air temperatures were rather low during our study period ranging from 0.005K to a single peak of 8.689K (ranging from daily means of 1.32K to 2.13K). As suggested by the reviewer, we provide some information on air-surface temperature differences over time in the attached Fig. 1.

The spatial differences of air-surface temperatures (Tmin and Tmax of the surface temperatures) extracted from the thermal maps are provided in the table below, averaged for the days of the year (DOY):

| DOY | Dif. LSTmin and AirTemp16.3 [K] | Dif. LSTmax and AirTemp16.3 [K] |
|---|---|---|
| 217 | 4.16 | 10.39 |
| 218 | 3.89 | 8.02 |
| 219 | 3.95 | 7.88 |
| 220 | 4.02 | 6.71 |
| 221 | 4.26 | 7.34 |

We will add the respective section to our manuscript in the results section:

*Temperature differences between measured air temperature at 16.3m (top of canopy) and mean land surface temperatures ranged from 0.005K to a single peak of 8.689K for the single flights while the daily average differences ranged from 1.32K to 2.13K.*

4. With 90% canopy cover, LST is mostly that of the leaf surfaces, i.e. reflecting the process of evaporative cooling of leaves by transpiration. Can the authors report these (evapo)transpiration values? A value is given in L360. Why are the units mm h-1 m-2? I thought that the mm already includes the area consideration (i.e., 1 mm = 1 L m-2).

We thank the reviewer for this insightful comment and agree that (evapo)transpiration should be provided in mm h-1. We will add more ET values to the respective section:

*At the time of the drone flights, eddy covariance-derived evapotranspiration was, on average, 0.43 $\pm$0.21 mm h$^{-1}$, with peak evapotranspiration of up to 0.87 mm h$^{-1}$ during midday.*

5. By using the EC data as absolute reference, the text seems to assume that the EC data are independently true. However, the EC is also an estimate based on an indirect

measurement. If there are any additional measurements that could further constrain these data, it would be very helpful. Regardless, the text should be adjusted to reflect that two estimates are compared, rather than an estimate to a direct measurement.

We thank the reviewer for this comment and fully agree. Since we used an errors-in-variables model (Deming regression) in our analysis, we did account for these measurement errors in both the x- and the y-axis (eddy covariance and drone-based method, respectively).

To further clarify this in the manuscript, we will add the following sentence to the statistics section (2.6):

*Both methods, the reference EC technique and the drone-based estimates, are associated with a certain degree of uncertainty. To account for the uncertainty in both, a model II Deming regression (Cornbleet and Gochman, 1979; Glaister, 2001) was applied for the analysis.*

6. In case that one doesn't have radiation measurements, would the DTD model be the best option to make use of the thermal information? In L400 the authors should note that such sensors must be tested independently in a separate study.

In case that no radiation measurements at all are available, the radiation budget can potentially be modelled according to location, date and time and under the assumption of cloud and haze free skies, which we tested in our study for all three models. However, these assumptions were frequently not met during our time of study, resulting in relatively poor net radiation estimates translating to inaccurate results for the DTD, TSEB-PT and DATTUTDUT model.

The reviewer also makes an important point regarding the testing of potential on-board sensor schemes. We will adjust the sentence accordingly:

*In our study, these measurements were taken with the EC equipment, but future stand-alone drone approaches are possible by using on-board miniaturized radiation sensors (Castro Aguilar et al., 2015; Suomalainen et al., 2018). However, the accuracy of such on-board radiation sensors should first be tested against reference methods, e.g. visually by scatter or inter-comparison plots (Castro Aguilar et al., 2015; Suomalainen et al., 2018) or with a model II regression procedure evaluating the interchangeability of methods and measurements (Passing and Bablok, 1983).*

7. The authors discuss measurements in drier sites. It would be interesting to compare these results with measurements of palm water-use and its effect on temperature. Below are a few studies on date palm, evidencing the high transpiration rates in a plantation, and the effect on temperature in an urban context.

We thank the reviewer for this interesting suggestion. The new drone-based method can likely help to link surface temperatures, e.g. in urban settings, and vegetation water use; however, this falls outside of the scope of the presented study. As mentioned before, we focus mainly on a method comparison rather than on applied ecological questions for now.

To clarify this further, we added a sentence to the discussion:

*Drone-based methods have a large untapped potential for ecological applications,*

*e.g. regarding ecohydrological optimization in land use systems and designing the climate-smart urban landscapes of the future.*

8. Finally, another potential comparison could be made with a study of transpiration of forest trees estimated by spatial temperature data from a thermal camera (see reference below).

Sperling, O., Shapira, O., Cohen, S., Tripler, E., Schwartz, A., & Lazarovitch, N. (2012). Estimating sap flux densities in date palm trees using the heat dissipation method and weighing lysimeters. Tree Physiology, 32(9), 1171-1178.

Potchter, O., Goldman, D., Kadish, D., & Iluz, D. (2008). The oasis effect in an extremely hot and arid climate: The case of southern Israel. Journal of Arid Environments, 72(9), 1721-1733.

Potchter, O., Goldman, D., Iluz, D., & Kadish, D. (2012). The climatic effect of a manmade oasis during winter season in a hyper arid zone: The case of Southern Israel. Journal of arid environments, 87, 231-242.

Lapidot, O., Ignat, T., Rud, R., Rog, I., Alchanatis, V., & Klein, T. (2019). Use of thermal imaging to detect evaporative cooling in coniferous and broadleaved tree species of the Mediterranean maquis. Agricultural and forest meteorology, 271, 285-294.

We thank the reviewer for this suggestion; as mentioned previously, this manuscript focuses on a method comparison rather than on the ecological application of the method and a comparison to other land-use types; in the (near) future, further work

will certainly also include further land-use types including old-growth and secondary tropical forest patches, agroforestry systems and smallholder plantations in lowland Sumatra and beyond.

We took up the reference suggested by the reviewer in the introduction:

*Transpiration from leaf surfaces leads to evaporative cooling of the canopy; LSTs, along with air temperature, can thus be used as a reliable indicator of plant water use, both in monocultures and in spatially highly heterogeneous systems such as natural forests (Lapidot et al., 2019).*

References:
Brenner, C., Thiem, C.E., Wizemann, H.-D., Bernhardt, M., Schulz, K., 2017. Estimating spatially distributed turbulent heat fluxes from high-resolution thermal imagery acquired with a UAV system. Int. J. Remote Sens. 38, 3003–3026. https://doi.org/10.1080/01431161.2017.1280202

Brenner, C., Zeeman, M., Bernhardt, M., Schulz, K., 2018. Estimation of evapo-transpiration of temperate grassland based on high-resolution thermal and visible range imagery from unmanned aerial systems. Int. J. Remote Sens. 39, 5141–5174. https://doi.org/10.1080/01431161.2018.1471550

Castro Aguilar, J.L., Gentle, A.R., Smith, G.B., Chen, D., 2015. A method to measure total atmospheric long-wave down-welling radiation using a low cost infrared ther-mometer tilted to the vertical. Energy 81, 233–244. https://doi.org/10.1016/j.energy.2014.12.035

Cornbleet, P.J., Gochman, N., 1979. Incorrect Least-Squares Regression Coefficients in Method- Comparison Analysis. Clin. Chem. 432–438.

Glaister, P., 2001. Least squares revisited. Math. Gaz. 85. https://doi.org/10.2307/3620485

Hoffmann, H., Nieto, H., Jensen, R., Guzinski, R., Zarco-Tejada, P., Friborg, T., 2016. Estimating evaporation with thermal UAV data and two-source energy balance models. Hydrol. Earth Syst. Sci. 20, 697–713. https://doi.org/10.5194/hess-20-697-2016

Lapidot, O., Ignat, T., Rud, R., Rog, I., Alchanatis, V., Klein, T., 2019. Use of thermal imaging to detect evaporative cooling in coniferous and broadleaved tree species of the Mediterranean maquis. Agric. For. Meteorol. 271, 285–294. https://doi.org/10.1016/j.agrformet.2019.02.014

Passing, H., Bablok, W., 1983. A new biometrical procedure for testing the equality of measurements from two different analytical methods. Application of linear regression procedures for method comparison studies in clinical chemistry, Part I. Clin. Chem. Lab. Med. 21. https://doi.org/10.1515/cclm.1983.21.11.709

Röll, A., Niu, F., Meijide, A., Ahongshangbam, J., Ehbrecht, M., Guillaume, T., Gunawan, D., Hardanto, A., Hendrayanto, Hertel, D., Kotowska, M.M., Kreft, H., Kuzyakov, Y., Leuschner, C., Nomura, M., Polle, A., Rembold, K., Sahner, J., Seidel, D., Zemp, D.C., Knohl, A., Hölscher, D., 2019. Transpiration on the rebound in lowland Sumatra. Agric. For. Meteorol. 274, 160–171. https://doi.org/10.1016/j.agrformet.2019.04.017

Sabajo, C.R., le Maire, G., June, T., Meijide, A., Roupsard, O., Knohl, A., 2017. Expansion of oil palm and other cash crops causes an increase of the land surface temperature in the Jambi province in Indonesia. Biogeosciences 14, 4619–4635. https://doi.org/10.5194/bg-14-4619-2017

Suomalainen, J., Hakala, T., Alves de Oliveira, R., Markelin, L., Viljanen, N., Näsi, R., Honkavaara, E., 2018. A Novel Tilt Correction Technique for Irradiance Sensors and Spectrometers On-Board Unmanned Aerial Vehicles. Remote Sens. 10, 2068. https://doi.org/10.3390/rs10122068

---

## Author Comment (AC2) · 13 Aug 2020

General comments The manuscript by Ellsasser et al. makes an interesting and useful contribution to the burgeoning literature on using UAVs to measure ecosystem properties and processes, in this case measurements of surface temperature for use in models of the surface energy balance to predict spatial variations in the latent heat flux and for comparison to eddy covariance-derived estimates of the same.

Dear Reviewer,

Thank you for taking the time to revise our manuscript. We welcome your comments and believe that they helped to improve our manuscript considerably. Please find our point-by point replies below.

Sincerely,
Florian Ellsäßer

The appendix describing the various energy balance/ET models should be better integrated with the main body of the manuscript, and as noted below some of the model equations need more clarification. In general, a reader should not have to read other previous papers to understand the approaches tested here (e.g., see my comments below regarding lines 174-175).

As suggested by the reviewer we will integrate the key information from the appendix into the main body of the manuscript starting from line 158:

*2.3 Energy balance models*

[revised manuscript text omitted]

I agree with the other reviewer that more discussion of the various uncertainties in EC-derived ET need to be discussed. While it is the reference method here it is also subject to many uncertainties.

As addressed in the reply for reviewer one, we will add the following information regarding uncertainties of the reference EC method:

Methods section:

*EC data processing and quality checks were performed following the methodology described in Meijide et al., (2017). Following Mauder and Foken, (2006), flux estimates during low turbulence and thus stable atmospheric conditions were removed from the analysis; however, low turbulence mainly occurred during night hours and was not observed during the daytime drone flights. Generally, the EC method is associated with uncertainties of 5 - 20% (Foken, 2008), mainly due to problems with energy balance closure (Wang et al. 2012). Further limitations are the high costs and quite specific requirements regarding size and terrain of the study site.*

Statistics section:

*Both methods, the reference EC technique and the drone-based estimates, are associated with a certain degree of uncertainty. To account for the uncertainty in both, a model II Deming regression (Deming, 1964) was applied for the analysis.*

The writing is generally fine but there are a few very awkward sentences that I suggest re-writing (see below).

We thank the reviewer for taking the time to point out the need for rewording these sentences. We revised them accordingly.
Specific comments

Lines 90-91: "the hottest and a group of coldest pixels in the image" – This is not and independent clause as it is missing a verb

*We will adjust the sentence accordingly:*

*In the one-source energy balance model DATTUTDUT (Deriving Atmosphere Turbulent Transport Useful To Dummies Using Temperature) (Timmermans et al., 2015) fluxes are estimated by relating single pixel temperatures to local temperature extremes.*

Lines 105-107: This sentence is confusing and needs to be re-written.

*We will adjust the sentence accordingly:*

*Model II method comparisons require a sample size of at least n=60 data pairs (Legendre and Legendre, 2003), which constrained previous studies with smaller sample sizes to using error terms and correlation coefficients instead of a full method comparison.*

Line 110: replace "presented" with "current"

*We will adjust the sentence accordingly:*

*The current study was conducted in the lowlands of Jambi province (Sumatra, Indone-*

*sia) where over the last decades, large areas of rainforest have been converted to rubber and oil palm plantations (Clough et al., 2016; Margono et al., 2012).*

Line 147: Quote the manufacturer's measurement uncertainty here, as you also discuss it later when mentioning thermal cameras. The true uncertainty is surely closer to 1-2 K for cameras like this.

As suggested by the reviewer, we will add more differentiated information on relative and absolute thermal accuracy to this section:

*The sensor covers spectral bands ranging from 7.5 to 13.5 $\mu$m with a relative thermal accuracy of 0.04 K and an absolute thermal accuracy of $\pm 2K$ (FLIR Systems, USA).*

Line 164: Provide the assumed surface emissivities used in each model and component

As suggested, we will add a sentence on assumed surface emissivities to the Methods:

*Since the DATTUTDUT model is a one-source energy balance model we used a uniform surface emissivity of 0.98 as recommended for vegetation dominated areas (Jones and Vaughan, 2010). For the two-source energy balance models we used a canopy emissivity of 0.98 and soil emissivity of 0.95. The emissivity values are based on averages for the 8-14 $\mu$m taken from Jones and Vaughan, (2010).*

Lines 174-175: Need to better explain this approach. P-T is usually used to predict LH fluxes not SH fluxes.

For the application of the TSEB-PT model we follow the workflow provided in Hoffmann et al., (2016). There, it is described in detail how the Priestley-Taylor (PT) approximation is used to calculate the canopy sensible heat flux from net radiation divergence estimates. This is now pointed out more clearly in the Methods of our manuscript:

*With $\triangle Rn$ known, sensible heat flux is then estimated using the Priestley-Taylor approximation.*

Lines 196-207: Do these models assume a closed energy balance? If so how does that affect your estimates?

As mentioned in the Methods section, all models assume energy balance closure; in accordance with the reference EC method, we applied the Bowen Ratio method for energy balance closure:

Line 228-230:

*As the applied drone-based models all assume full energy balance closure, we used the Bowen ratio closure method (Pan et al., 2017; Twine et al., 2000) to compute full closure for the EC measurements. The Bowen ratio method was found to produce the most congruent results in conjunction with drone-based latent heat flux estimates (Brenner et al., 2017) and was therefore applied in this study.*

Line 219: Was this an aspirated measurement of Tair?

We appreciate this insightful question by the reviewer. We originally used the Tair measurements at 22m on the EC tower but, inspired by the reviewer's comment, have re-run all models with the temperature measurements at 16.3 m (i.e. ∼2m above the canopy). However, the absolute average temperature difference between the two measurement heights is below 0.24 °C.

We have adjusted the following sentence in the methods section:

Air temperature and relative humidity were measured with thermohygrometers (type 1.1025.55.000, Thies Clima, Göttingen, Germany) at 16.3 m height. We re-ran the models with the temperature measurements at 16.3 m. We further received an email with recommendations on how to improve the model performance (e.g. vegetation parameters) of the two-source energy balance models and started implementing these in the models. The revised manuscript will also include the fully revised model results. The pre-results are saved in the supplementary material.

Line 222: These are IRTs not thermal cameras, so you do not know exactly which canopy elements you are measuring! Were they capturing only leaves all of the time? Also, what surface emissivity was assumed for these measurements of surface temperature? Did you correct for the influences of reflected longwave radiation, relative humidity, distance to object, etc? And what are measurement uncertainties of the IRTs?

We thank the reviewer for this valuable comment. To avoid confusion, we now consistently apply the term IRT throughout the manuscript. We will further add more detail on the issues raised by the reviewer to the Methods:

*The two IRTs used in our study (IR100 Radiometer, Campbell Scientific Inc., Logan, USA) have a field-of-view (FOV) of 8-10°. Considering the distance from their fixed location on the tower to the average height of the oil palm canopy, they cover a circular area of 2.2 m², over which they average the received thermal signal. The recorded canopy area comprises different functional parts of the canopy (e.g. leaflets, petioles). On average, we assumed a surface emissivity of 0.98 for the canopy area (Jones and Vaughan, 2010). We did not correct the values recorded with the IRTs for any other influences; the distance from the canopy surface to the sensors was only about 10-12m.*

Line 229: Describe the Bowen ratio closure method in more detail.

As suggested by the reviewer, we will add more detail about the Bowen ratio closure method to the Methods:

*The energy balance closure (EBC) of the reference EC measurements was 0.77 ($r^2$ = 0.87), which is in line with EBC reported for other tall vegetation canopies (Stoy et al., 2013). Since the used energy balance models assume full EBC, we applied the so-called Bowen ratio closure method to the EC data (Pan et al., 2017). The method assumes that wind measurements miss some of the total covariance and dispersive fluxes. Therefore, underestimations of LE and H are carried over proportionally because of similarity among fluxes (Twine et al., 2000). The Bowen ratio closure method proportionally assigns the underestimated turbulent energy to LE and H fluxes to reach full EBC.*

Line 247: "systematic"

We will adjust the sentence accordingly:

*Statistics such as $r^2$ have their limitations in method comparison since they are designed to indicate how well the resulting model of the regression describes the outcome and are not necessarily a good measure for systematic bias between methods.*

Line 273: I think you mean "alive"

We will adjust the sentence accordingly:

*The plantation is very well managed, so that all oil palm canopies are alive, no oil palms have died and only dry leaves are removed.*

Lines 278-286: As noted above these measurements were not made with thermal camera but with IRTs. Please update.

As mentioned above we will now consistently apply the term IRTs throughout the manuscript.

Line 280: Is the 122 number based on 2 maps/flight?

Yes. We re-worded the sentence to point this out more clearly:

*To check whether the two IRTs measure similar temperatures compared to drone*

*recorded LSTs, we extracted a total of 122 'IRT-sized' (i.e. ~2.2 m$^2$) LST footprints from the drone-recorded maps and plotted the measured and adjusted temperatures of both recording systems against each other (Fig. A1).*

Line 293-294: Is this peak SW measured during the flight or average SW?

We applied 10 min averages of all SW data that were recorded during a respective flight. We will add this information to the methods section:

*The measured solar-irradiance was recorded as 10 min averages of short-wave irradiance.*

Line 295: By "canopy air temperature" do you mean the Tair measured at 22m?

We thank the reviewer for this valuable question. As already mentioned in a previous reply, we originally used Tair as measured at 22m on the EC tower, but now have re-run all models using Tair measured at 16.3m (i.e. ~ 2m above the canopy).

We have adjusted the following sentence in the methods section:

*Air temperature and relative humidity were measured with thermohygrometers (type 1.1025.55.000, Thies Clima, Göttingen, Germany) at 16.3 m height.*

Line 302-303: This is an awkward sentence – rewrite.

As suggested, we re-wrote the sentence:

*Congruence of LE estimates with reference EC measurements differed among the three applied models and was further affected by the configuration of the Rn assessment (Fig. A3).*

Line 303: The first time you cite Fig. A3 you need to discuss why modeled Rnet is so poor.

As suggested by the reviewer we will add a sentence discussing the poor performance when applying modelled Rnet:

*The assumptions for Rn_mod were not always met as cloud cover was present during several flights (Table A1); consequently, the corresponding net radiation estimates were too high, leading to a substantial overestimation of latent heat fluxes.*

Line 304: Replace "congruence" with "agreement" or "fidelity"

We will adjust the sentence accordingly:

*Generally, error metrics were reduced and agreement was increased the more measurement-controlled the Rn determination process was.*

Line 307-308: Perhaps this poor agreement in morning and late afternoon is not surprising since the dATTUDUT method is based on modeled Rnet..?

We thank the reviewer for this insightful comment. We will add a section to the manuscript that addresses both this comment and the following comment (please refer to the following answer).

Line 308-309: It's worth breaking out the description of the performance of the TSEBPT estimates into a separate sentence. Are these estimates uniformly higher than the EC estimates or only during part of the day?

We thank the reviewer for this insightful comment. We will add a section to the manuscript that addresses both this comment and the previous comment:

*DATTUTDUT LE estimates closely agreed with EC measurements around noon, but were higher in the morning and afternoon hours, which is caused by overestimations of Rn from the Rn_mod method. LE estimates from TSEB-PT were consistently higher than EC measurements, with particularly large divergences around noon (Fig. 2a).*

Lines 335-336: Seems like this sentence is missing a word or two.

We adjusted the sentence accordingly:

*The TSEB-PT model in Rn_mes configuration also showed no significant continuous errors but was subject to proportional bias (Fig. 4c). The TSEB-PT overestimated LE particularly around noon, when fluxes are very high (Fig. 2c and 3c).*

Line 352: I'm unclear what you mean about the X-level for the bias in EC reference fluxes.

The bias of two applied methods can be expressed in an X- and a Y-level, bias on X-axis (horizontal) and bias on Y-axis (vertical) respectively. We were particularly interested in the bias of the new drone-based methods based on the EC technique (here: X-level).

Lines 405-406: Are you referring to the slope in this sentence?

We agree with the reviewer that the wording was previously unprecise and adjusted the sentence accordingly:

*An opposite situation was found for the DTD model where the confidence intervals for the slope indicated no proportional errors, but the intercept revealed a continuous error for the analytical method.*

Lines 455-457: Well before this discussion of errors you should define what you mean by proportional versus continuous errors.

Following the suggestion by the reviewer, we will add the following information to the Methods section:

*If the confidence intervals for the intercept of the Deming regression include zero, there is no constant or continuous error between the two methods. If the confidence intervals for the intercept do not include zero, both methods differ by a constant amount, i.e. the new method has a continuous error compared to the reference method. In contrast, the confidence intervals of the slope of the Deming regression indicate whether there*

*is a proportional error between the methods, which increases proportionally with the magnitude of the predicted value.*

Line 500: Replace "results in" with "predict"

We will adjust the sentence accordingly:

*In the histograms of LE fluxes from all pixels within the single studied footprints (Fig. 6), the DATTUTDUT and DTD models predict a bell-shaped normal distribution but very different value ranges.*

Line 503: eliminate comma after "both"

We will adjust the sentence accordingly:

*Mean and median are very similar indicating close to zero skewness for both the DATTUTDUT and DTD model.*

Line 520: Which edge? Computer or edge of study area?

We adjusted the sentence for further clarification:

*Autonomous acquisition of LSTs over EC stations and the surrounding areas can be supplemented by on-board and ground sensors. Energy-balance models can then potentially be calculated using edge computing schemes on-board the drone to enable*

*a dense temporal resolution of LST, flux and ET maps in almost real-time.*

Line 542: Replace "cameras" with "IRTs"

As mentioned above, we will now consistently apply the term IRTs throughout the manuscript.

Line 565: How are the surface epsilon (emissivity) terms estimated? Do they vary spatially across the image?

We thank the reviewer for this insightful comment, and have expanded the according method section to clarify the issues raised by the reviewer. As already mentioned above, we will add this information to the methods part of the manuscript.

*Since the DATTUTDUT model is a one-source energy balance model we used a uniform surface emissivity of 0.98 as recommended for vegetation dominated areas (Jones and Vaughan, 2010). For the two-source energy balance models we used a canopy emissivity of 0.98 and soil emissivity of 0.95. The emissivity values are based on averages for the 8-14 $\mu$m taken from Jones and Vaughan, (2010).*

Lines 579-580: Show the equations for calculating radiometric LSTs.

Since we used a radiometric thermal camera we did not have to calculate the radiometric LSTs from a greyscale picture (as e.g. in Cohen et al., 2005); there thus is no equation. The energy-balance models in our study use the directional radiometric temperature that was recorded with the thermal camera on the drone. A further

substitution of temperatures or correction procedures (e.g. excess resistance) is not necessary (Hoffmann et al., 2016). We will add a sentence to the Methods to point this out more clearly:

*For the TSEB-PT and DTD model directional radiometric temperatures are used and no calculation of aerodynamic temperature by using an excess resistance term is needed (Hoffmann et al., 2016). The proximity of the thermal camera to the surface is much closer compared to other typical carriers (such as satellites or planes) and hence atmospheric effects are supposed to be largely reduced. To use a uniform input for all the applied models, we used directional radiometric temperature recordings from the drone as input without applying further corrections.*

Line 588: I assume this (Po) is a shortwave albedo?

Correct, Po is the short-wave surface albedo. It was taken from Timmermans et al., (2015). We added a sentence to the Methods to clarify this:

*Surface albedo P0 is calculated as in Timmermans et al. (2015) based on the assumption that dense vegetation appears colder than rocks or soil in the thermal imagery (Brutsaert, 1982; Garratt, 1992).*

Line 600: This model assumes cloud-free conditions (with a constant transmissivity)?

Yes, in its original formulation the DATTUTDUT model assumes cloud free conditions (Timmermans et al., 2015). For simplicity Timmermans et al. (2015) suggest using a constant value of 0.7 for the transmissivity or to follow a simple parameterization

scheme for instantaneous shortwave atmospheric transmissivity following the description in Burridge and Gadd, (1977). We chose the second option and calculated short-wave transmissivity using the solar elevation angle. We added a sentence to the Methods to point this out more clearly:

*Transmissivity $\tau$ is calculated as described in Burridge and Gadd, (1977) using the solar elevation angle $\alpha$ that was determined from the geographic position of our site and the coordinated universal time (UTC) of the measurements.*

Line 605: Is that supposed to be an epsilon symbol as in equation 2?

Yes, this is supposed to be a $\epsilon$atm. We have adapted the manuscript accordingly:

*Timmermans et al. (2015) suggest using a constant value of 0.7 for $\tau$ and 0.8 atmospheric emissivity ($\epsilon$atm), but as our flight times range from 09:00 to 16:30 h local time we decided to include the solar elevation angle as in eq. 5.*

[revised manuscript text omitted]

**Supplement:**

DATTUTDUT          TSEB-PT          DTD

(a) Models with fully modelled net radiation (Rn_mod)

$y = 341 + 0.25\ x$
$r^2 = 0.24\quad p < .001$
$MAE = 125$
$RMSE = 147$

$y = 555 + 0.10\ x$
$r^2 = 0.03\quad p = 0.18$
$MAE = 252$
$RMSE = 287$

$y = 560 + 0.22\ x$
$r^2 = 0.31\quad p < .001$
$MAE = 291$
$RMSE = 315$

(b) Models with net radiation from measured short-wave irradiance (Rn_sw)

$y = -318 + 1.87\ x$
$r^2 = 0.31\quad p < .001$
$MAE = 122$
$RMSE = 165$

$y = -282 + 2.14\ x$
$r^2 = 0.3\quad p < .001$
$MAE = 151$
$RMSE = 204$

$y = -163 + 1.92\ x$
$r^2 = 0.37\quad p < .001$
$MAE = 170$
$RMSE = 218$

(c) Models with measured net radiation (Rn_mes)

$y = -31 + 1.14\ x$
$r^2 = 0.85\quad p < .001$
$MAE = 47$
$RMSE = 60$

$y = -47 + 1.55\ x$
$r^2 = 0.77\quad p < .001$
$MAE = 158$
$RMSE = 174$

$y = 6 + 1.53\ x$
$r^2 = 0.82\quad p < .001$
$MAE = 190$
$RMSE = 206$

Latent heat flux - thermal imagery [W m$^{-2}$]

Latent heat flux - eddy covariance [W m$^{-2}$]

---

## Author Response (AR1)

Dear Prof. Yakir, please find enclosed our revised manuscript *'Predicting evapotranspiration from drone-based thermography – a method comparison in a tropical oil palm plantation'*. We thank you and the reviewers for the provided comments and suggestions, which we all considered in our revised manuscript. They are addressed in detail in the enclosed point-by-point replies. We apologize for the delays in revising the manuscript, which occurred in the wake of the lead author's doctoral thesis submission and defense and subsequent move to a new job and university. We included the suggestions and revised the manuscript accordingly. We hope that the improvements are convincing to you and the reviewers so that the manuscript can move on to publication in Biogeosciences.

Sincerely,

Florian Ellsäßer on behalf of all co-authors

**1 Response to reviewer I**

Dear Reviewer,

Thank you for taking the time to revise our manuscript. We welcome your comments and think they have helped to improve our manuscript considerably. Please find our point-by point response (in blue color) below.

Sincerely,
Florian Ellsäßer

Peer review for the manuscript: Predicting evapotranspiration from drone-based thermography – a method comparison in a tropical oil palm plantation by Ellsasser et al The manuscript under consideration reports a 9-days study of surface temperature measurements over an oil palm plantation in Indonesia using a thermal camera mounted on a drone. The authors used the temperature data to calculate the latent heat flux using three different models, with/out radiation inputs, and showed good agreement between one of the models and the latent heat estimated from an eddy-covariance (EC) calculation based on an on-site flux tower. The drone-based temperature calculation is more flexible than the EC, also providing spatial information at high resolution. This is a very nice paper reporting an elegant study. The text and figures are carefully prepared and nicely presented.

I have only a few questions and suggestions:

1. Considering the rather narrow variation in air temperature over the tropical plantation, would you think that the fact that the study was performed at this site is a challenge? Or rather an easier case? I think that this point is touched upon, but further discussion would be appreciated.

We agree with the reviewer that there was rather narrow variation during the time of study (canopy air temperature ranged from 22.5 to 32.3 °C), as is typical for the region. Generally, the study site was rather challenging. We added a short section taking up these points to the discussion (L605-L611):

*Generally, the equatorial study site was rather challenging due to high temperatures and humidity and frequent occurrence of haze, as well as for logistical reasons. Additionally, many previous drone-based studies were conducted on grasslands (e.g. Brenner et al. (2017, 2018)) or on low-growing crops such as wheat fields (Hoffmann et al., 2016), but not on crops with a rather complex canopy structure such as oil palm. On the other hand, our study site showed large temperature differences between soil and canopy, which simplified the distinguishing of each fraction.*

2. Considering the aggravating situation of deforestation in the studied region, and the implications on surface warming (L110-113), it would be highly interesting to make a comparison study between the palm plantation and the natural rainforest. I assume that the higher spatial heterogeneity in the latter would offer a better test case for the spatial distribution of ET (Fig. 5). Can the authors include such information?

We thank the reviewer for this very interesting point. Indeed the comparison of land surface temperatures and modelled evapotranspiration of natural rainforest and an oil palm plantations would provide valuable spatial insight into the current transformation of transpiration patterns caused by local- and regional-scale land-use changes, as e.g. described in Röll et al., 2019 and Sabajo et al., 2017. However, the present study focuses on the comparison of different drone-based methods as a baseline for future ecological studies, rather than applying the methods to different land-use types. We will however follow up on this in the future, as we also performed flight missions over flooded and non-flooded natural forest sites and a variety of adjacent areas including mixed oil palm stands and small holder rubber and oil palm plantations.

To clarify this point in the manuscript, we updated the introduction section with the following sentence (L124-L125):

*The present study focuses on the comparison of different drone-based methods as a baseline for future ecological studies, rather than applying the methods to different land-use types.*

3. It would be good to include in the paper some information on the measured air-surface temperature differences as function of time and space.

The differences of mean land surface and air temperatures were rather low during our study period ranging from 0.005K to a single peak of 8.689K and daily means ranged from daily means of 1.32K to 2.13K. The following figure provides an overview of the air-surface temperature differences over the study period in the Fig. 1 below:

[Figure]

The spatial differences of air-surface temperatures (Tmin and Tmax of the surface temperatures) extracted from the thermal maps are provided in the table below, averaged for the days of the year (DOY):

| DOY | Dif. LSTmin and AirTemp16.3 [K] | Dif. LSTmax and AirTemp16.3 [K] |
|-----|--------------------------------|--------------------------------|
| 217 | 4.16 | 10.39 |
| 218 | 3.89 | 8.02 |
| 219 | 3.95 | 7.88 |
| 220 | 4.02 | 6.71 |
| 221 | 4.26 | 7.34 |

As suggested by the reviewer, we added a sentence summarizing this information to the Results (L418-L420):

*Temperature differences between measured air temperature at 16.3m (top of canopy) and mean land surface temperatures ranged from 0.005K to a single peak of 8.689K for the single flights while the daily averaged differences ranged from 1.32K to 2.13K.*

4. With 90% canopy cover, LST is mostly that of the leaf surfaces, i.e. reflecting the process of evaporative cooling of leaves by transpiration. Can the authors report these (evapo)transpiration values? A value is given in L360. Why are the units mm h-1 m-2? I thought that the mm already includes the area consideration (i.e., 1 mm = 1 L m-2).

We thank the reviewer for this insightful comment and agree that (evapo)transpiration should be provided in mm $h^{-1}$. We added more ET values to the respective section (L425-L427):

*At the time of the drone flights, LE from the EC method ranged between 87 and 596 W $m^{-2}$ (mean: 337 W $m^{-2}$) and eddy covariance-derived evapotranspiration was on average, 0.43 $\pm$0.21 mm $h^{-1}$, with peak evapotranspiration of up to 0.87 mm $h^{-1}$ during midday.*

5. By using the EC data as absolute reference, the text seems to assume that the EC data are independently true. However, the EC is also an estimate based on an indirect measurement. If there are any additional measurements that could further constrain these data, it would be very helpful. Regardless, the text should be adjusted to reflect that two estimates are compared, rather than an estimate to a direct measurement.

We thank the reviewer for this comment and fully agree. Since we used an errors-in-variables model (Deming regression) in our analysis, we did account for these measurement errors in both the x- and the y-axis (eddy covariance and drone-based method, respectively).

To further clarify this in the manuscript, we added the following sentence to the statistics section (L356-L359):

*Both methods, the reference EC technique and the drone-based estimates, are associated with a certain degree of uncertainty. To account for the uncertainty in both, a model II Deming regression (Deming, 1964) was applied for the analysis to consider uncertainties in both x and y variables (Cornbleet and Gochman, 1979; Glaister, 2001).*

6. In case that one doesn't have radiation measurements, would the DTD model be the best option to make use of the thermal information? In L400 the authors should note that such sensors must be tested independently in a separate study.

In case that no radiation measurements at all are available, the radiation budget can potentially be modelled according to location, date and time and under the assumption of cloud and haze free skies, which we tested in our study for all three models. However, these assumptions were frequently not met during our time of study, resulting in relatively poor net radiation estimates translating to inaccurate results for the DTD, TSEB-PT and DATTUTDUT model.

The reviewer also makes an important point regarding the testing of potential on-board sensor schemes. We adjusted the sentence accordingly (L540-L546):

*In our study, these measurements were taken with the EC equipment, but future stand-alone drone approaches are possible by using on-board miniaturized radiation sensors (Castro Aguilar et al., 2015; Suomalainen et al., 2018). However, the accuracy of such on-board radiation sensors should first be tested against reference methods, e.g. visually by scatter or inter-comparison plots (Castro Aguilar et al., 2015; Suomalainen et al., 2018) or with a model II regression procedure evaluating the interchangeability of methods and measurements (Passing and Bablok, 1983).*

7. The authors discuss measurements in drier sites. It would be interesting to compare these results with measurements of palm water-use and its effect on temperature. Below are a few studies on date palm, evidencing the high transpiration rates in a plantation, and the effect on temperature in an urban context.

We thank the reviewer for this interesting suggestion. The new drone-based method can likely help to link surface temperatures, e.g. in urban settings, and vegetation water use; however, this falls outside of the scope of the presented study. As mentioned before, we focus mainly on a method comparison rather than on applied ecological questions for now.

To clarify this further, we added a sentence to the discussion (L644-L646):

*Drone-based methods have a large untapped potential for ecological applications, e.g. regarding ecohydrological optimization in land use systems and designing the climate-smart urban landscapes of the future.*

8. Finally, another potential comparison could be made with a study of transpiration of forest trees estimated by spatial temperature data from a thermal camera (see reference below).

Sperling, O., Shapira, O., Cohen, S., Tripler, E., Schwartz, A., & Lazarovitch, N. (2012). Estimating sap flux densities in date palm trees using the heat dissipation method and weighing lysimeters. Tree Physiology, 32(9), 1171-1178.

Potchter, O., Goldman, D., Kadish, D., & Iluz, D. (2008). The oasis effect in an extremely hot and arid climate: The case of southern Israel. Journal of Arid Environments, 72(9), 1721-1733.

Potchter, O., Goldman, D., Iluz, D., & Kadish, D. (2012). The climatic effect of a manmade oasis during winter season in a hyper arid zone: The case of Southern Israel. Journal of arid environments, 87, 231-242.

Lapidot, O., Ignat, T., Rud, R., Rog, I., Alchanatis, V., & Klein, T. (2019). Use of thermal imaging to detect evaporative cooling in coniferous and broadleaved tree species of the Mediterranean maquis. Agricultural and forest meteorology, 271, 285-294.

We thank the reviewer for this suggestion; as mentioned previously, this manuscript focuses on a method comparison rather than on the ecological application of the method and a comparison to other land-use types; in the (near) future, further work will certainly also include further land-use types including old-growth and secondary tropical forest patches, agroforestry systems and smallholder plantations in lowland Sumatra and beyond.

We took up the reference suggested by the reviewer in the introduction (L59-L61):

*Transpiration from leaf surfaces leads to evaporative cooling of the canopy; LSTs, along with air temperature, can thus be used as a reliable indicator of plant water use, both in monocultures and in spatially highly heterogeneous systems such as natural forests (Lapidot et al., 2019).*

**2 Response to reviewer II**

Dear Reviewer,

Thank you for taking the time to revise our manuscript. We welcome your comments and believe that they helped to improve our manuscript considerably. Please find our point-by point replies below.

Sincerely,
Florian Ellsäßer

General comments The manuscript by Ellsasser et al. makes an interesting and useful contribution to the burgeoning literature on using UAVs to measure ecosystem properties and processes, in this case measurements of surface temperature for use in models of the surface energy balance to predict spatial variations in the latent heat flux and for comparison to eddy covariance-derived estimates of the same.

The appendix describing the various energy balance/ET models should be better integrated with the main body of the manuscript, and as noted below some of the model equations need more clarification. In general, a reader should not have to read other previous papers to understand the approaches tested here (e.g., see my comments below regarding lines 174-175).

As suggested by the reviewer we integrated the key information from the appendix into the main body of the manuscript (L164-L308):

*2.3 Energy balance models*

[revised manuscript text omitted]

I agree with the other reviewer that more discussion of the various uncertainties in EC-derived ET need to be discussed. While it is the reference method here it is also subject to many uncertainties.

As addressed in the reply to reviewer one, we added the following information regarding uncertainties of the reference EC method:

Methods section (L347-L352):

*EC data processing and quality checks were performed following the methodology described in (Meijide et al., 2017). Following (Mauder and Foken, 2006), flux estimates during low turbulence and thus stable atmospheric conditions were removed from the analysis; however, low turbulence mainly occurred during night hours and was not observed during the daytime drone flights. Generally, the EC method is associated with uncertainties of 5 - 20% (Foken, 2008). Further limitations are the high costs and quite specific requirements regarding size and terrain of the study site.*

Statistics section (L356-L359):

*Both methods, the reference EC technique and the drone-based estimates, are associated with a certain degree of uncertainty. To account for the uncertainty in both, a model II Deming regression (Deming, 1964) was applied for the analysis to consider uncertainties in both x and y variables (Cornbleet and Gochman, 1979; Glaister, 2001).*

The writing is generally fine but there are a few very awkward sentences that I suggest re-writing (see below).

We thank the reviewer for taking the time to point out the need for rewording these sentences. We revised them accordingly.

Specific comments

Lines 90-91: "the hottest and a group of coldest pixels in the image" – This is not and independent clause as it is missing a verb

We adjusted the sentence accordingly (L91-L93):

*In the one-source energy balance model DATTUTDUT (Deriving Atmosphere Turbulent Transport Useful To Dummies Using Temperature) (Timmermans et al., 2015) fluxes are estimated by relating single pixel temperatures to local temperature extremes.*

Lines 105-107: This sentence is confusing and needs to be re-written.

We adjusted the sentence accordingly (L109-L112):

*Since full method comparisons based on model II regression require a sample size of at least n=60 data pairs (Legendre and Legendre, 2003), many previous studies with smaller sample sizes were constrained to using error terms and correlation coefficients.*

Line 110: replace "presented" with "current"

We adjusted the sentence accordingly (L114-L116):

*The current study was conducted in the lowlands of Jambi province (Sumatra, Indonesia) where over the last decades, large areas of rainforest have been converted to rubber and oil palm plantations (Clough et al., 2016; Margono et al., 2012).*

Line 147: Quote the manufacturer's measurement uncertainty here, as you also discuss it later when mentioning thermal cameras. The true uncertainty is surely closer to 1-2 K for cameras like this.

As suggested by the reviewer, we added more differentiated information on relative and absolute thermal accuracy to this section (L151-L152):

*The sensor covers spectral bands ranging from 7.5 to 13.5 µm with a relative thermal accuracy of 0.04 K and an absolute thermal accuracy of $\pm 2K$ (FLIR Systems, USA).*

Line 164: Provide the assumed surface emissivities used in each model and component

As suggested, we added a sentence on assumed surface emissivities to the Methods (L216-L218):

*Further, we used a constant surface emissivity ($\epsilon_{surf}$) of 0.98 as recommended for vegetation dominated areas (Jones and Vaughan, 2010) and not 1.0 as simplified in the original formulation of the DATTUTDUT model.*

and (L249-L250):

*For the two-source energy balance models we used a canopy emissivity of 0.98 and soil emissivity of 0.95. The emissivity values are based on averages for the 8-14 µm taken from Jones and Vaughan, (2010).*

Lines 174-175: Need to better explain this approach. P-T is usually used to predict LH fluxes not SH fluxes.

For the application of the TSEB-PT model we follow the workflow provided in Hoffmann et al., (2016). There, it is described in detail how the Priestley-Taylor (PT) approximation is used to calculate the canopy sensible heat flux from net radiation divergence estimates. This is now pointed out more clearly in the Methods of our manuscript (L266-L267):

*With $\Delta Rn$ known, sensible heat flux is then estimated using the Priestley-Taylor approximation following the approach by Hoffmann et al., (2016).*

Lines 196-207: Do these models assume a closed energy balance? If so how does that affect your estimates?

As mentioned in the Methods section, all models assume energy balance closure; in accordance with the reference EC method, we applied the Bowen Ratio method for energy balance closure (L335-L346):

*As the applied drone-based models all assume full energy balance closure, we used the Bowen ratio closure method (Pan et al., 2017; Twine et al., 2000) to compute full closure for the EC measurements. The Bowen ratio method was found to produce the most congruent results in conjunction with drone-based latent heat flux estimates (Brenner et al., 2017) and was therefore applied in this study. The energy balance closure (EBC) of the reference EC measurements was 0.77 ($r^2 = 0.87$), which is in line with EBC reported for other tall vegetation canopies (Stoy et al., 2013). Since the used energy balance models assume full EBC, we applied the so-called Bowen ratio closure method to the EC data (Pan et al., 2017). The method assumes that wind measurements miss some of the total covariance and dispersive fluxes. Therefore, underestimations of LE and H are carried over proportionally because of similarity among fluxes (Twine et al., 2000). The Bowen ratio closure method proportionally assigns the underestimated turbulent energy to LE and H fluxes to reach full EBC.*

Line 219: Was this an aspirated measurement of Tair?

We appreciate this insightful question by the reviewer. We originally used the Tair measurements at 22m on the EC tower but, inspired by the reviewer's comment, have re-run all models with the temperature measurements at 16.3 m (i.e. ∼2m above the canopy). However, the absolute average temperature difference between the two measurement heights is below 0.24 °C.

We have adjusted the following sentence in the methods section (L319-L321):

*Air temperature and relative humidity were measured with thermohygrometers (type 1.1025.55.000, Thies Clima, Göttingen, Germany) at 16.3 m height.*

We re-ran the models with the temperature measurements at 16.3 m. We further received an email with recommendations on how to improve the model performance (e.g. vegetation parameters) of the two-source energy balance models and implementd these in the models. The revised manuscript includes these fully revised models, as shown in the key figure below:

[Figure]

Line 222: These are IRTs not thermal cameras, so you do not know exactly which canopy elements you are measuring! Were they capturing only leaves all of the time? Also, what surface emissivity was assumed for these measurements of surface temperature? Did you correct for the influences of reflected longwave radiation, relative humidity, distance to object, etc? And what are measurement uncertainties of the IRTs?

We thank the reviewer for this valuable comment. To avoid confusion, we now consistently apply the term IRT throughout the manuscript. We further added more detail on the issues raised by the reviewer to the Methods (L323-L330):

*The two IRTs used in our study (IR100 Radiometer, Campbell Scientific Inc., Logan, USA) have a field-of-view (FOV) of 8-10°. Considering the distance from*

*their fixed location on the tower to the average height of the oil palm canopy, they cover a circular area of 2.2 $m^2$, over which they average the received thermal signal. The recorded canopy area comprises different functional parts of the canopy (e.g. leaflets, petioles). On average, we assumed a surface emissivity of 0.98 for the canopy area (Jones and Vaughan, 2010). We did not correct the values recorded with the IRTs for any other influences; the distance from the canopy surface to the sensors was only about 10m.*

Line 229: Describe the Bowen ratio closure method in more detail.

As suggested by the reviewer, we added more detail about the Bowen ratio closure method to the Methods (L338-L344):

*The energy balance closure (EBC) of the reference EC measurements was 0.77 ($r^2 = 0.87$), which is in line with EBC reported for other tall vegetation canopies (Stoy et al., 2013). Since the used energy balance models assume full EBC, we applied the so-called Bowen ratio closure method to the EC data (Pan et al., 2017). The method assumes that wind measurements miss some of the total covariance and dispersive fluxes. Therefore, underestimations of LE and H are carried over proportionally because of similarity among fluxes (Twine et al., 2000). The Bowen ratio closure method proportionally assigns the underestimated turbulent energy to LE and H fluxes to reach full EBC.*

Line 247: "systematic"

We adjusted the sentence accordingly (L367-L369):

*Statistics such as $r^2$ have their limitations in method comparison since they are designed to indicate how well the resulting model of the regression describes the outcome and are not necessarily a good measure for systematic bias between methods.*

Line 273: I think you mean "alive"

We adjusted the sentence accordingly (L391-L392):

*The plantation is very well managed, so that all oil palm canopies are alive, no oil palms have died and only dry leaves are removed.*

Lines 278-286: As noted above these measurements were not made with thermal camera but with IRTs. Please update.

As mentioned above we now consistently apply the term IRTs throughout the manuscript.

Line 280: Is the 122 number based on 2 maps/flight?

Yes. We re-worded the sentence to point this out more clearly (L398-L402):

*To check whether the two IRTs measure similar temperatures compared to drone recorded LSTs, we extracted a total of 122 'IRT-sized' (i.e. $\sim$2.2 $m^2$) LST footprints from the drone-recorded maps. A correlation of both temperature measurements revealed a small deviation of the measured temperatures resulting in a mean absolute error (MAE) and root mean squared error (RMSE) of 1.59 and 2.15 K respectively.*

Line 293-294: Is this peak SW measured during the flight or average SW?

We applied 10 min averages of all SW data that were recorded during a respective flight. We added this information to the discussion section (L550-L552):

*The short-wave irradiance measurements used in this study were stored as 10 min averages that probably didn't represent the high level of irradiance variations in the tropical study area adequately.*

Line 295: By "canopy air temperature" do you mean the Tair measured at 22m?

We thank the reviewer for this valuable question. As already mentioned in a previous response, we originally used Tair as measured at 22m on the EC tower, but now have re-run all models using Tair measured at 16.3m (i.e. $\sim$ 2m above the canopy).

We have adjusted the following sentence in the methods section (L319-L320):

*Air temperature and relative humidity were measured with thermohygrometers (type 1.1025.55.000, Thies Clima, Göttingen, Germany) at 16.3 m height.*

Line 302-303: This is an awkward sentence – rewrite.

As suggested, we re-wrote the sentence (L425-LL427):

*Congruence of LE estimates with reference EC measurements differed among the three applied models and was further affected by the configuration of the Rn assessment (Fig. 2).*

Line 303: The first time you cite Fig. A3 you need to discuss why modeled Rnet is so poor.

As suggested by the reviewer we added a short section discussing the poor performance when applying modelled Rnet (L427-L433):

*The assumptions for Rn_mod were not always met as cloud cover was present during several flights; consequently, the corresponding net radiation estimates were too high, leading to a substantial overestimation especially of smaller latent heat fluxes. The short-wave irradiance based Rn_sw configuration resulted in Rn estimates that were by average very comparable with the measured net radiation Rn_mes but also showed a rather high variation (Fig. 2). Generally, error metrics were reduced and agreement was increased the more measurement-controlled the Rn determination process was.*

Line 304: Replace "congruence" with "agreement" or "fidelity"

We adjusted the sentence accordingly (L431-L433):

*Generally, error metrics were reduced and agreement was increased the more measurement-controlled the Rn determination process was.*

Line 307-308: Perhaps this poor agreement in morning and late afternoon is not surprising since the dATTUDUT method is based on modeled Rnet..?

We thank the reviewer for this insightful comment. We added a section to the manuscript that addresses both this comment and the following comment (please refer to the following answer).

Line 308-309: It's worth breaking out the description of the performance of the TSEBPT estimates into a separate sentence. Are these estimates uniformly higher than the EC estimates or only during part of the day?

We thank the reviewer for this insightful comment. We added a section to the manuscript that addresses both this comment and the previous comment (L437-L442):

*DATTUTDUT LE estimates closely agreed with EC measurements around noon, but were higher in the morning and afternoon hours, which is caused by overestimations of Rn from the Rn_mod method (Fig. 3a). LE estimates from TSEB-PT were consistently higher than EC measurements, with particularly large divergences around noon (Fig. 3a). The LE predictions from the DTD model in Rn_mod configuration were rather overestimated, especially around noon when compared with the EC reference measurements (Fig. 3a).*

Lines 335-336: Seems like this sentence is missing a word or two.

We adjusted the sentence accordingly (L474-L477):

*The TSEB-PT model in Rn_mes configuration also showed no significant continuous errors but was subject to a minor proportional bias (Fig. 5c). The TSEB-PT model overestimated LE particularly around noon, when fluxes are very high (Fig. 3c and 4c).*

Line 352: I'm unclear what you mean about the X-level for the bias in EC reference fluxes.

The bias of two applied methods can be expressed in an X- and a Y-level, bias on X-axis (horizontal) and bias on Y-axis (vertical) respectively. We were particularly interested in the bias of the new drone-based methods based on the EC technique (here: X-level).

Lines 405-406: Are you referring to the slope in this sentence?

We agree with the reviewer that the wording was previously unprecise and adjusted the sentence accordingly (L477-L478):

*The DTD model also showed no continuous bias but indicated a proportional error in the analytical method and the Jackknife method (Fig. 5c).*

Lines 455-457: Well before this discussion of errors you should define what you mean by proportional versus continuous errors.

Following the suggestion by the reviewer, we added the following information to the Methods section (L465-L470):

*If the confidence intervals for the intercept of the Deming regression include zero, there is no constant or continuous error between the two methods. If the confidence intervals for the intercept do not include zero, both methods differ by a constant amount, i.e. the new method has a continuous error compared to the reference method. In contrast, the confidence intervals of the slope of the Deming regression indicate whether there is a proportional error between the methods, which increases proportionally with the magnitude of the predicted value.*

Line 500: Replace "results in" with "predict"

We changed the sentence (L507-L509):

*Both distributions of the two-source energy balance models show gaps in the histogram, while the histogram of the DATTUTDUT model displays a more con-*

*tinuous distribution (Fig. 7)*

Line 503: eliminate comma after "both"

We adjusted the sentence accordingly (L634-L635):

*For the DATTUTDUT model mean and median are very similar indicating close to zero skewness.*

Line 520: Which edge? Computer or edge of study area?

We adjusted the sentence for further clarification (L647-L650):

*Autonomous acquisition of LSTs over EC stations and the surrounding area scan be supplemented by on-board and ground sensors. Energy-balance models can then potentially be calculated using edge computing schemes on-board the drone to enable a dense temporal resolution of LST, flux and ET maps in almost real-time.*

Line 542: Replace "cameras" with "IRTs"

As mentioned above, we will now consistently apply the term IRTs throughout the manuscript.

Line 565: How are the surface epsilon (emissivity) terms estimated? Do they vary spatially across the image?

We thank the reviewer for this insightful comment, and have expanded the according method section to clarify the issues raised by the reviewer. As already mentioned above, we will add this information to the methods part of the manuscript (L216-L218).

*Further, we used a constant surface emissivity ($\epsilon_{surf}$) of 0.98 as recommended for vegetation dominated areas (Jones and Vaughan, 2010) and not 1.0 as simplified in the original formulation of the DATTUTDUT model.*

Lines 579-580: Show the equations for calculating radiometric LSTs.

Since we used a radiometric thermal camera we did not have to calculate the radiometric LSTs from a greyscale picture (as e.g. in Cohen et al., 2005); there thus is no equation. The energy-balance models in our study use the directional radiometric temperature that was recorded with the thermal camera on the drone. A further substitution of temperatures or correction procedures (e.g. excess resistance) is not necessary (Hoffmann et al., 2016). We will add a sentence to the Methods to point this out more clearly:

*For the TSEB-PT and DTD model directional radiometric temperatures are used and no calculation of aerodynamic temperature by using an excess resistance term is needed (Hoffmann et al., 2016). The proximity of the thermal camera to the surface is much closer compared to other typical carriers (such as satellites or planes) and hence atmospheric effects are supposed to be largely reduced. To use a uniform input for all the applied models, we used directional radiometric temperature recordings from the drone as input without applying further corrections.*

Line 588: I assume this (Po) is a shortwave albedo?

Correct, Po is the short-wave surface albedo. It was taken from Timmermans et al., (2015). We added a sentence to the Methods to clarify this (L199-L200):

*Surface albedo P0 is calculated as in Timmermans et al. (2015) based on the assumption that dense vegetation appears colder than rocks or soil in the thermal imagery (Brutsaert, 1982; Garratt, 1992).*

Line 600: This model assumes cloud-free conditions (with a constant transmissivity)?

Yes, in its original formulation the DATTUTDUT model assumes cloud free conditions (Timmermans et al., 2015). For simplicity Timmermans et al. (2015) suggest using a constant value of 0.7 for the transmissivity or to follow a simple parameterization scheme for instantaneous shortwave atmospheric transmissivity following the description in Burridge and Gadd, (1977). We chose the second option and calculated short-wave transmissivity using the solar elevation angle. We added a sentence to the Methods to point this out more clearly (L206-L208):

*Transmissivity $\tau$ is calculated as described in Burridge and Gadd, (1977) using the solar elevation angle $\alpha$ that was determined from the geographic position of our site and the coordinated universal time (UTC) of the measurements.*

Line 605: Is that supposed to be an epsilon symbol as in equation 2?

Yes, this is supposed to be a $\epsilon$atm. We have adapted the manuscript accordingly (L214-L216):

*Timmermans et al. (2015) suggest using a constant value of 0.7 for $\tau$ and 0.8 atmospheric emissivity ($\epsilon$atm), but as our flight times range from 09:00 to 16:30 h local time we decided to include the solar elevation angle as in eq. 5.*

---

## Author Response (AR2)

Dear Prof. Yakir,

please find enclosed our manuscript 'Predicting evapotranspiration from drone-based thermography – a method comparison in a tropical oil palm plantation' with the required technical corrections. We address all changes in the point-by-point replies below.

Sincerely,

Florian Ellsäßer on behalf of all co-authors

This paper report in the application of thermal radiation drone measurements to estimate evapotranspiration from the canopy of oil-palm plantation. It received supporting reviews from two expert Reviewers but with an extensive list of comments, which required revisions. The authors have now responded in detail to all the comments and revised the paper accordingly. Carefully checking the paper and Reviewers comments I find that all comments were related to clarifications, additional detail, and in-text corrections, and therefore consider it as minor revisions that do not require additional time investment of the reviewers. I therefore recommend publication in the parent form.

I indicated the need for technical corrections that do not need reviewing, as I would ask the authors to consider indicating around eq. 2 that the outgoing radiation is subtracted,

*We added a section that clarifies this aspect to the manuscript (L 187-193):*

*Where the short-wave component is calculated by multiplying incoming short-wave radiation $R_s\downarrow$ [W $m^{-2}$] with its absorption ratio deducted from the combined soil and vegetation albedo $\alpha$. This way, reflected outgoing short-wave radiation $R_s\uparrow$ is subtracted from the energy balance. The long-wave radiation budget is calculated from surface (soil and vegetation) emissivity $\varepsilon_{surf}$ and atmospheric emissivity $\varepsilon_{atm}$, the Stefan-Boltzmann constant $\sigma$ (5.6704*10^{-8} W $m^{-2}$*$K^{-4}$), air temperature $T_{air}$ and radiometric land surface temperature $T(\theta)_{surf}$ (both in K). The incoming long-wave radiation component is added to the budget and the outgoing long-wave radiation component is subtracted.*

and consider the inclusion of comments on potential limitations, such as due to significant effects of clouds on atmospheric emissivity, or of wind and turbulence cooling competing with evaporative cooling, when introducing a method for general use.

*We thank the editor for this suggestion and added a section to the Discussion part of the manuscript (L588-592):*

*Limitations of the presented methods compared with the reference EC method however still exist. As such, the thermography-based recording process for land surface temperatures can be affected by water vapor, haze or dust, which increase atmospheric emissivity. Also, wind and turbulence cooling effects that compete with evaporative cooling are not captured in this approach.*